# RIM-binding proteins recruit BK-channels to presynaptic release sites adjacent to voltage-gated Ca²⁺-channels

Alessandra Sclip[1],[*] , Claudio Acuna[1,2], Fujun Luo[1,3] & Thomas C Südhof[1],[**]

## Abstract

The active zone of presynaptic nerve terminals organizes the neurotransmitter release machinery, thereby enabling fast Ca²⁺-triggered synaptic vesicle exocytosis. BK-channels are Ca²⁺-activated large-conductance K⁺-channels that require close proximity to Ca²⁺-channels for activation and control Ca²⁺-triggered neurotransmitter release by accelerating membrane repolarization during action potential firing. How BK-channels are recruited to presynaptic Ca²⁺-channels, however, is unknown. Here, we show that RBPs (for RIM-binding proteins), which are evolutionarily conserved active zone proteins containing SH3- and FN3-domains, directly bind to BK-channels. We find that RBPs interact with RIMs and Ca²⁺-channels via their SH3-domains, but to BK-channels via their FN3-domains. Deletion of RBPs in calyx of Held synapses decreased and decelerated presynaptic BK-currents and depleted BK-channels from active zones. Our data suggest that RBPs recruit BK-channels into a RIM-based macromolecular active zone complex that includes Ca²⁺-channels, synaptic vesicles, and the membrane fusion machinery, thereby enabling tight spatio-temporal coupling of Ca²⁺-influx to Ca²⁺-triggered neurotransmitter release in a presynaptic terminal.

**Keywords** active zone; BK-channels; nano-domain coupling; neurotransmitter release; voltage-gated calcium channels
**Subject Categories** Membrane & Intracellular Transport; Neuroscience; Physiology
The EMBO Journal (2018) 37: e98637

## Introduction

At the active zone of presynaptic terminals, action potentials (APs) cause voltage-gated Ca²⁺-channels to open. The resulting transient increase in cytoplasmic Ca²⁺ induces neurotransmitter release by triggering synaptic vesicle exocytosis (Walter *et al*, 2011; Sudhof, 2013). The number and location of voltage-gated Ca²⁺-channels at the active zone and the coupling of voltage-gated Ca²⁺-channels to synaptic vesicles are critical in determining the strength and plasticity of synapses (Meinrenken *et al*, 2002; Modchang *et al*, 2010; Eggermann *et al*, 2011). Several mechanisms control the size and duration of the Ca²⁺-signals at the active zone, thereby generating temporally restricted micro- and nano-domains of Ca²⁺ that determine the extent of release (e.g., see Fedchyshyn & Wang, 2005; Bucurenciu *et al*, 2008; Vyleta & Jonas, 2014). Among the most important of these mechanisms is the control of the AP duration by BK-channels (Fakler & Adelman, 2008; Contet *et al*, 2016; Griguoli *et al*, 2016). BK-channels are Ca²⁺-activated large-conductance K⁺-channels that limit the duration of an AP, thereby controlling the extent of Ca²⁺-channel opening and of neurotransmitter release per AP. BK-channels are enriched in presynaptic terminals and are present in a complex with voltage-gated Ca²⁺-channels, but it is unknown how BK-channels are recruited to presynaptic terminals and how they are molecularly connected with voltage-gated Ca²⁺-channels (Roberts, 1993; Berkefeld *et al*, 2006; Indriati *et al*, 2013).

Voltage-gated Ca²⁺-channels are localized to active zones by binding to a multi-domain protein complex composed of RIMs and RIM-binding proteins (RBPs); this complex also directly interacts with synaptic vesicles and the exocytotic machinery and forms the core of the active zone (Deng *et al*, 2011; Han *et al*, 2011; Kaeser *et al*, 2011; Liu *et al*, 2011; Davydova *et al*, 2014; Acuna *et al*, 2015, 2016; Muller *et al*, 2015; Grauel *et al*, 2016; reviewed in Sudhof, 2012). In mammals, RBPs are encoded by three genes: RBP1 and RBP2 are differentially expressed throughout the brain, while RBP3 is primarily expressed outside the brain (Wang *et al*, 2000; Mittelstaedt & Schoch, 2007). RIM-binding proteins are composed of three SH3-domains and three fibronectin-type III domains (FN3-domains) that are flanked by variable interspersed sequences. The SH3-domains of RBPs directly bind to proline-rich sequences in RIMs and in L-, and P/Q- and N-type voltage-gated Ca²⁺-channels (Wang *et al*, 2000; Hibino *et al*, 2002; Kaeser *et al*, 2011). Via these interactions, RBPs contribute to the presynaptic recruitment of voltage-gated Ca²⁺-channels at *Drosophila* neuromuscular junctions (Liu *et al*, 2011), of N- and P/Q-type Ca²⁺-channels at standard chemical synapses (Acuna *et al*, 2015, 2016; Grauel *et al*, 2016),

1 Department of Cellular and Molecular Physiology, Howard Hughes Medical Institute, Stanford University School of Medicine, Stanford, CA, USA
2 CH Schaller Foundation and Institute of Anatomy and Cell Biology, Heidelberg University, Heidelberg, Germany
3 School of Life Sciences, Guangzhou University, Guangzhou, China
*Corresponding author. Tel: +1 650 721 1418; E-mail: asclip@stanford.edu
**Corresponding author. Tel: +1 650 721 1418; Fax: +1 650 498 4585; E-mail: tcs1@stanford.edu

and of L-type $Ca^{2+}$-channels at ribbon synapses (Krinner et al, 2017; Luo et al, 2017). RIMs interact with $Ca^{2+}$-channels both indirectly via binding to RBPs and directly via their PDZ-domain that bind to the C-terminus of N- and P/Q-type $Ca^{2+}$-channels (Kaeser et al, 2011). In addition, RIMs interact with Rab3-proteins on synaptic vesicles, with Bassoon in the cytomatrix, and with Munc13 in the vesicle fusion machinery (Wang et al, 1997, 2000; Betz et al, 2001; Schoch et al, 2002; Deng et al, 2011; Kaeser et al, 2011; Davydova et al, 2014). Thus, RIMs nucleate a macromolecular complex that includes $Ca^{2+}$-channels, synaptic vesicles, and the release machinery in addition to RBPs (Sudhof, 2012).

At standard chemical mammalian synapses in which N- and P/Q-type $Ca^{2+}$-channels mediate presynaptic $Ca^{2+}$-influx, RIMs are the primary drivers for synaptic recruitment of $Ca^{2+}$-channels (Kaeser et al, 2011), whereas RBPs support RIMs in organizing nano-scale coupling of $Ca^{2+}$-influx to release (Acuna et al, 2015; Grauel et al, 2016). As a result, at these synapses (as measured at the calyx of Held synapse), deletion of RIMs massively decreases presynaptic $Ca^{2+}$-currents (Han et al, 2011; Kaeser et al, 2011), deletion of RBPs decreases the fidelity of neurotransmitter release without changing overall $Ca^{2+}$-currents (Acuna et al, 2015), and deletion of both RIMs and RBPs nearly abolishes presynaptic $Ca^{2+}$-currents (Acuna et al, 2016). At ribbon synapses that use L-type voltage-gated $Ca^{2+}$-channels, however, RBPs are primarily responsible for recruiting voltage-gated $Ca^{2+}$-channels because RBPs but not RIMs bind to L-type $Ca^{2+}$-channels (Hibino et al, 2002; Kaeser et al, 2011). Here, deletion of RBPs has a much more pronounced effect on release at ribbon synapses than at standard chemical synapses (Luo et al, 2017).

BK-channels are formed by tetramers of BKα-subunits that have seven transmembrane regions and a long cytoplasmic sequence containing two regulator-of-$K^+$-conductance (RCK) domains (the RCK1- and RCK2-domains; Atkinson et al, 1991; Adelman et al, 1992; Butler et al, 1993; Lee & Cui, 2010). The RCK-domains regulate gating of BK-channels via $Ca^{2+}$-binding, phosphorylation, and interactions with other proteins (Xia et al, 2002; Yusifov et al, 2008; Yuan et al, 2010). BKα proteins associate with cytoplasmic auxiliary BKβ- and BKγ-subunits that affect the gating properties of BK-channels and may contribute to their subcellular targeting (Meera et al, 1996; Tseng-Crank et al, 1996; Wallner et al, 1996; Orio et al, 2002). BK-channels are ubiquitously expressed in the brain (Sausbier et al, 2006) and enriched in presynaptic terminals (Robitaille et al, 1993a; Knaus et al, 1996; Zhou et al, 1999; Hu et al, 2001; Ishikawa et al, 2003; Misonou et al, 2006; Nakamura & Takahashi, 2007), where they form a molecular complex with $Ca^{2+}$-channels, and are thus optimally located for coupling $Ca^{2+}$-influx to the $Ca^{2+}$-regulation of the duration of an AP (Roberts, 1993; Robitaille et al, 1993b; Berkefeld et al, 2006; Loane et al, 2007; Wang, 2008).

High $Ca^{2+}$-concentrations (~10 μM) are required for activation of BK-channels, implying that BK-channels are physiologically activated only in close proximity to voltage-gated $Ca^{2+}$-channels and preferentially during AP trains (Berkefeld & Fakler, 2013). As a result, it has been suggested that nano-domain coupling of BK-channels to voltage-gated $Ca^{2+}$-channels is necessary to trigger opening of BK-channels during APs (Berkefeld et al, 2006; Fakler & Adelman, 2008). Selective block of BK-channels with iberiotoxin (IbTX) or paxilline increased the probability of glutamate release in CA3-CA3 synapses (Raffaelli et al, 2004) and regulated neurotransmitter

release in A17 amacrine cells in the retina (Grimes et al, 2009). Moreover, at the calyx of Held synapse (Ishikawa et al, 2003; Nakamura & Takahashi, 2007), in hippocampal mossy fibre synapses (Alle et al, 2011), and at CA3-CA1 Schaffer collateral synapses (Hu et al, 2001), a presynaptic function of BK-channels in regulating neurotransmitter release was uncovered when voltage-gated $K^+$-channels were blocked with 4-aminopyridine. Overall, the current evidence suggests that BK-channels are an important mechanism of regulating neurotransmitter release especially during AP trains, but the molecular interactions that recruit BK-channels to presynaptic terminals are unclear.

Here, we identified BK-channels in an unbiased yeast two-hybrid screen as novel interaction partner of RBPs. The binding of RBPs to BK-channels required the FN3-domains of RBPs and the RCK-domains of BK-channels. These findings led us to propose that RBPs function as scaffolds at the release site, and that by simultaneously binding to BK-channels (through the FN3-domains) and voltage-gated $Ca^{2+}$-channels and RIMs (through their SH3-domains), RBPs contribute to the nano-domain coupling of $Ca^{2+}$-influx to $Ca^{2+}$-triggered synaptic vesicle exocytosis. Our results thus describe a function for the conserved FN3-domains of RBPs and suggest a mechanism by which BK-channels are recruited to voltage-gated $Ca^{2+}$-channels at the release site in the active zone.

# Results

## RBPs bind to BKα-channels

RBP1 and RBP2 are large multi-domain proteins that contain one N-terminal SH3-domain, three central FN3-domains, and two C-terminal SH3-domains (Fig 1A; Mittelstaedt & Schoch, 2007; Wang et al, 2000). To identify new binding partners for RBPs, we performed unbiased large-scale yeast two-hybrid screens using three fragments of rat RBP2 as baits: a fragment (residues 247–859) containing the FN3-domains and the linker region; a fragment (residues 1–859) containing the N-terminal SH3-domains, the FN3-domains, and the long central linker sequence; and a fragment (residues 247–1,068) containing the FN3-domains, linker region, and C-terminal SH3-domains (Fig 1A, Appendix Fig S1A). We isolated and re-confirmed a total of 71 prey clones (Appendix Fig S1B, Tables S1 and S2). Four of the prey clones that were isolated with the RBP2-247-859 bait encoded the cytoplasmic RCK2-domain of BKα-channels (Fig 1A, Appendix Table S1). Moreover, as previously described (Hibino et al, 2002; Kaeser et al, 2011), multiple clones representing the L-type $Ca^{2+}$-channel α-subunit (Cacna1b) were recovered with baits containing the SH3-domains, validating the overall screen (Appendix Fig S1B and Table S1). In addition, several other preys of potential interest were recovered (Appendix Table S1), but were not pursued further because of the potential importance of a direct interaction of RBPs with BKα-channels.

To independently assess the interaction between RBP2 and BKα-channels, we performed co-immunoprecipitation experiments with proteins expressed in HEK293T cells (Fig 1B and C). We co-expressed myc-tagged full-length RBP2 with the YFP-tagged BKα RCK2-domain that we had isolated in the yeast two-hybrid screen. As negative and positive controls, respectively, we used YFP alone

**Figure 1.  RBP2 interacts with the α-subunit of BK-channels.**

A    Diagrams of RBP2 protein structure and bait construct used for the yeast two-hybrid screen which leads to the discovery of the interaction between RBP2 and BK channels (above), and location of the BKα prey sequences in the BKα domain structure (below; not depicted in scale). For details, see Appendix Fig S1, Table S1 and S2.

B    Experimental strategy for validating the interaction of the RBP2 FN3-domains with BKα RCK-domains using co-immunoprecipitations (see C) or for imaging experiments (see D and E) on transfected HEK293T cells expressing various combinations of RBP2 and BKα proteins.

C    Co-immunoprecipitation experiments to test the interaction of RBP2 with BKα. Cell lysates from HEK293T cells expressing YFP, YFP-tagged full-length RIM1α, or the YFP-tagged RCK2-domain of BKα (from prey clone 38) either alone or together with myc-tagged full-length RBP2 were subjected to immunoprecipitations with antibodies to GFP (which recognize YFP; left) or RBP2 (right). Input fractions (In; 1% of total) and immunoprecipitates (IP) were analysed by immunoblotting with antibodies to the myc-epitope (red; top) or GFP (green; bottom). Bands were visualized with fluorescently labelled secondary antibodies (B, negative control; arrows label specific YFP-positive bands).

D    Imaging of transfected HEK293T cells expressing YFP alone or the YFP-tagged BKα RCK2-domain without or with myc-tagged full-length RBP2 (red, myc-epitope; blue, DAPI; green, YFP). Top, representative images (scale bar, 5 μm); bottom, fluorescence line scans across the cells to visualize the relative locations of YFP, DAPI, and myc signals (intensity was normalized to the maximal point and is expressed in arbitrary units). Note that under overexpression conditions, YFP and the YFP-tagged BKα RCK2-domain in the absence of RBP2 is partly nuclear, but YFP-tagged BKα is selectively recruited to the plasma membrane by RBP2, providing an imaging assay of the binding of BKα to RBP2.

E    Quantitative analysis in transfected HEK293T cells of the co-localization of YFP (left) or the YFP-tagged BKα RCK2-domain (right) with the DAPI-stained nucleus as a function of the co-expression of RBP2. Graphs show the cumulative distribution (left) and the mean (right) of the Pearson correlation coefficient between the YFP and DAPI signals. Note that RBP2 dramatically reduces the correlation coefficient as a measure of the recruitment of the BKα RCK2-domain out of the nucleus to the cell membrane. Bar graphs show means ± SEM; *n* (cells/experiments): YFP only = 18/4, YFP + RBP2 = 17/4, YFP-BKα = 26/4, and YFP-BKα + RBP2 = 25/4. Statistical significance was calculated using Student's *t*-test (***$P < 0.001$).

F, G    Co-immunoprecipitation of endogenous BKα and RBP2 from mouse brain homogenates (F, experimental strategy; G, representative immunoblots). Brain proteins solubilized with Triton X-100 (1%), NP-40 (1%), or SDS (0.3%) were immunoprecipitated with antibodies to RBP2. The input fraction (In; 1% of total protein) and immunoprecipitates (IP) as well as a negative control (B; preimmune serum) were analysed by immunoblotting using antibodies to RBP2 (4193) or BKα (APC-021). For more extensive studies, see Appendix Fig S2.

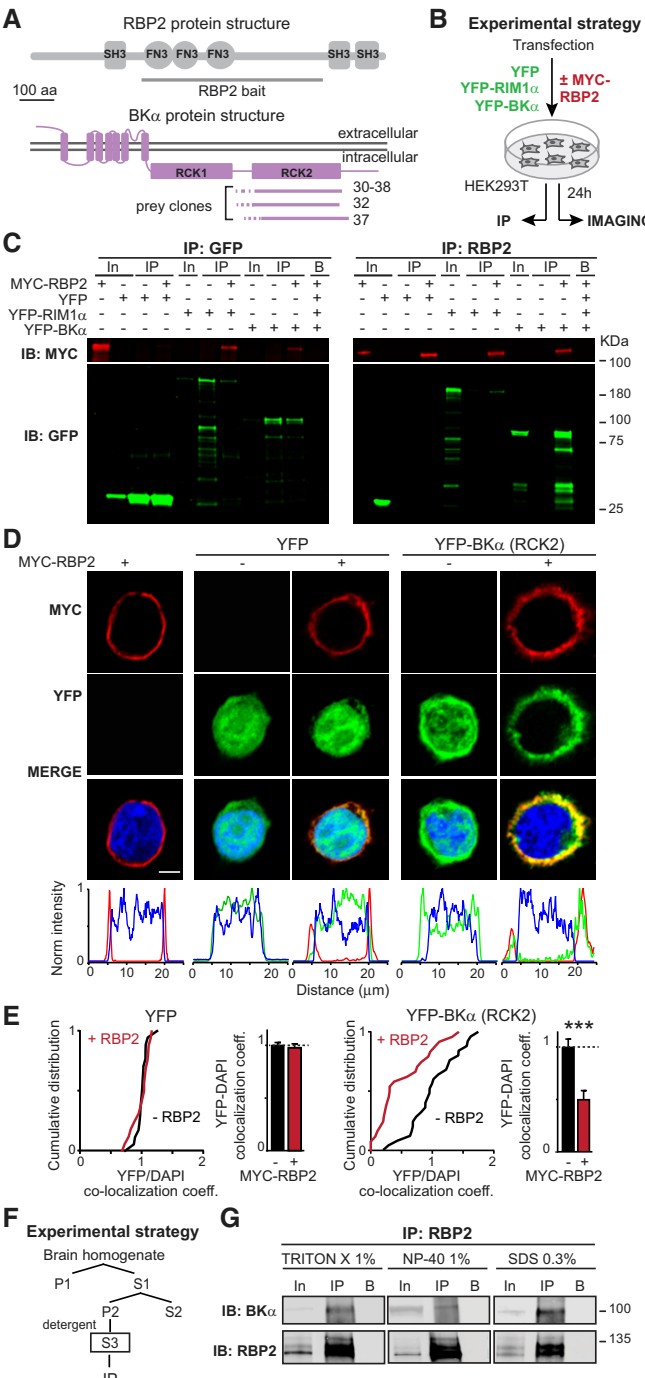

or YFP-tagged full-length RIM1α (Fig 1B). We then confirmed by co-immunoprecipitations with GFP antibodies (which recognize YFP) or with RBP2 antibodies that myc-tagged RBP2 can bind to both RIM and BKα-channels, whereas YFP alone could not bind to either molecule (Fig 1C).

Next, we examined the interaction between RBP2 and BKα-channels by imaging their localizations in transfected HEK293T cells (Fig 1D and E). We expressed the YFP-tagged RCK2-domain of BKα alone or together with myc-tagged full-length RBP2 in HEK293T cells, immunostained the cells with myc-specific antibodies, and then quantified fluorescence signals relative to the nucleus, using DAPI as a marker of nuclei. The YFP-tagged RCK2-domain of BKα and YFP, when expressed alone, was localized to all cellular compartments (as is often observed for overexpressed proteins), whereas RBP2 expressed alone quantitatively localized

to the plasma membrane (Fig 1D). Co-expression of RBP2 with the RCK2-domain, however, caused the translocation of the RCK2-domain to the membrane, confirming binding (Fig 1D). Co-expression of RBP2 with YFP, which is also localized in all cellular compartments, had no effect. Quantifications using correlation analyses confirmed that YFP-tagged RCK2-domain and YFP co-localized with the nuclear DAPI-stain in the absence of RBP2, and that YFP-tagged RCK2-domain but not YFP were recruited to the plasma membrane in the presence of RBP2, thus revealing their interaction in the context of the expressing cell (Fig 1E).

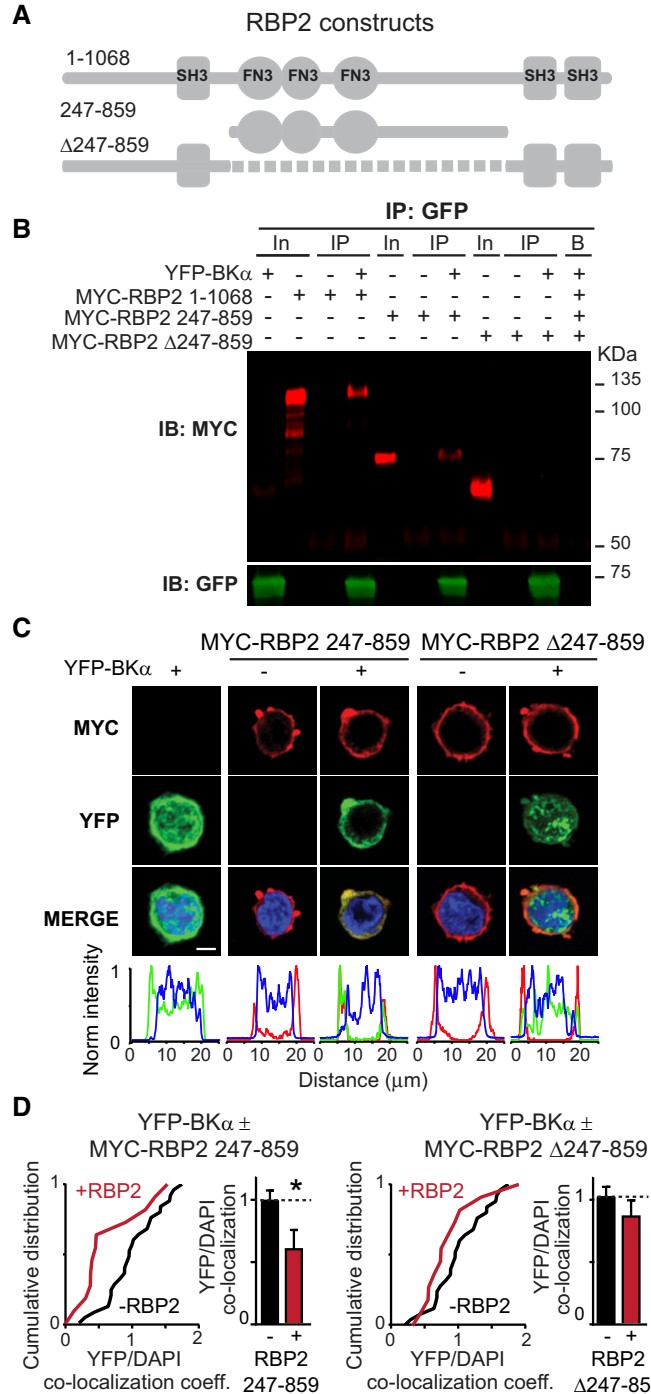

**Figure 2. Interaction of RBP2 with BKα requires RBP2 FN3-domains.**

A  Domain structures of myc-tagged RBP2 constructs tested for interactions with BKα (from top: full-length RBP2 [residues 1–1,068], RBP2 sequence containing the FN3-domains [residues 247–859], and N- and C-terminal RBP2 sequences lacking the central FN3-domains [Δ247–859]).

B  YFP-immunoprecipitations of proteins from HEK293T cells expressing YFP-tagged BKα alone, or the three myc-tagged RBP2 constructs alone or together with YFP-BKα. Input fractions (In; 1% of total) or immunoprecipitates (IP) were immunoblotted with antibodies to myc (red) or GFP (green, recognizes YFP). Note that RBP2 is only immunoprecipitated in the presence of YFP-BKα.

C  Representative images (top; scale bar, 5 µm) and line scans (bottom) of HEK293T cells expressing myc-tagged RBP2 [247–859] or RBP2 [Δ247–859] without or with YFP-tagged BKα. Cells were stained for myc (red), DAPI (blue), or GFP (green). Line scans compare the localizations of myc, DAPI, and YFP.

D  Cumulative plots and bar graphs of the correlation coefficients between the YFP-tagged BKα and the nuclear DAPI signals in HEK293T cells expressing myc-RBP2 (247–859) (top) or myc-RBP2 (Δ247–859) (bottom) proteins without or with YFP-BKα. Bar graphs show means ± SEM; *n* (cells/experiments): YFP-BKα = 26/4, YFP-BKα + RBP2 247–859 = 12/3, and YFP-BKα + RBP2 Δ247–859 = 12/3. Statistical significance was calculated using Student's *t*-test (*$P < 0.05$).

to explore a variety of detergents for the immunoprecipitations, including low concentrations of SDS. We found that in these immunoprecipitations, BKα was co-enriched with both RBP2 and RIMs, whereas syntaxin-1 (used as a negative control) was not (Fig 1G, Appendix Fig S2B). These results are consistent with the presence of a complex composed of endogenous BKα, RBP2, and RIMs. Importantly, no signal for BKα was detected when immunoprecipitations were performed with samples from constitutive RBP1,2 double KO mice, confirming the specificity of the interaction (Appendix Fig S2C).

## RBP sequences containing FN3-domains bind to both RCK-domains of BKα-channels

To map the domains of RBP2 that bind to BKα-channels, we co-expressed different fragments of myc-tagged RBP2 with YFP-tagged BKα-channels in HEK293T cells and tested binding by immunoprecipitation and imaging (Fig 2A). As assayed by co-immunoprecipitations, full-length RBP2 (aa 1–1,068) as expected bound robustly to BKα-protein (Fig 2B). A truncated version of RBP2 containing the FN3-domains and adjacent linker sequences (aa 247–859) bound reproducibly but weakly to BKα-channels, whereas a RBP2 construct containing a deletion of the FN3-domains and linker sequences (Δ247–859) failed to bind (Fig 2B). The same binding relationships were observed when measured by imaging the RBP2-dependent translocation of the YFP-tagged RCK2-domain of BKα out of the nucleus (Fig 2C and D). These results confirm the yeast two-hybrid experiment and suggest that the FN3-domains of RBPs are required for RBP binding to BKα-channels.

We next examined the region of BKα-channels that mediates BKα-binding to RBPs. The intracellular sequences of BKα-channels are composed of tandem RCK-domains, the RCK1-domain (aa 391–692) and RCK2-domain (aa 761–1,178), that are homologous to each other (Appendix Fig S3A and B). We therefore tested if the interaction of RBP2 with the RCK2-domain, as shown above, is

## Endogenous BKα and RBP2 form a complex in the brain

The experiments up to now establish an *in vitro* interaction of RBP2 with the RCK2-domain of BKα. To test whether endogenous RBP2 and BKα-channels form a complex *in vivo*, we immunoprecipitated RBP2 or RIMs from brain homogenates, and immunoblotted the immunoprecipitated proteins for BKα, RBP2, and RIMs (Fig 1F and G, Appendix Fig S2). The challenge inherent to these experiments is that RIMs and RBPs are part of the active zone protein complex that is largely detergent-insoluble, prompting us

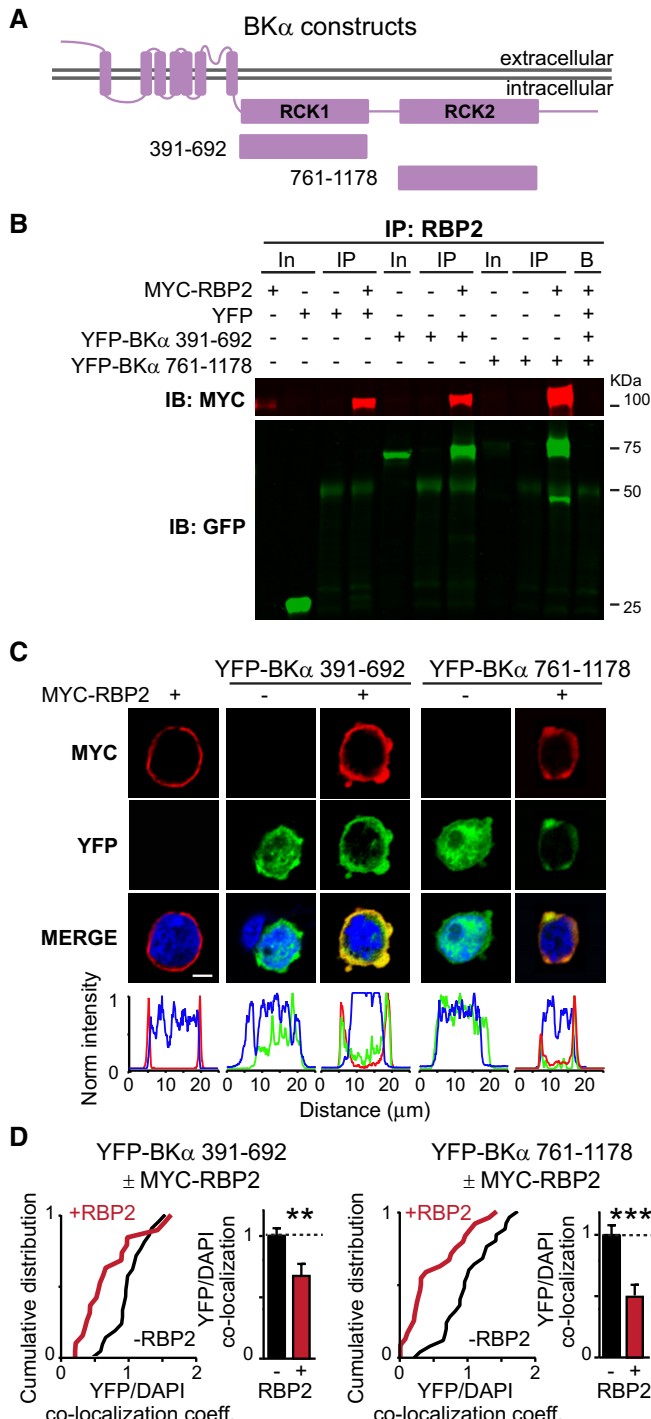

**B**

IP: RBP2

**Figure 3. The RCK1 or RCK2 domain of BKα both bind to RBP2.**

A  Domain structures of full-length BKα (top) and of the two BKα fragments containing the RCK1 [residues 391–692] or RCK2 domain [residues 761–1,178] that were tested for RBP2 interactions.

B  Myc-immunoprecipitations of cell lysates from transfected HEK293T cells expressing either myc-tagged RBP2 alone or together with YFP or the indicated YFP-tagged BKα RCK-domains. Input fractions (In; 1% of total) or immunoprecipitates (IP) were analysed by immunoblotting with antibodies to myc (red) or GFP (green, recognizes YFP). For more experiments, see Appendix Fig S3.

C  Representative images (top; scale bar, 5 μm) or line scans (bottom) of HEK293T cells expressing myc-tagged RBP2 alone, or the YFP-tagged BKα RCK1 or RCK2 domains alone or together with myc-tagged RBP2. Cells were stained for the myc-epitope (red) or DAPI (blue). Note that the BKα RCK-domains (green YFP fluorescence) are soluble when expressed alone, but are recruited to the cell membrane when co-expressed with RBP2. See also Fig 1D (experiments shown in Figs 1D and 3C were done in the same batch).

D  Cumulative plots and bar graphs of the correlation coefficients between YFP-tagged BKα RCK1 (top) and RCK2 domains (bottom) and nuclear DAPI with or without co-expression of RBP2. Bar graphs show means ± SEM; *n* (cells/experiments) = YFP-BKα 391–692 = 26/4, YFP-BKα 391–692 + full-length RBP2 = 25/4, YFP-BKα 761–1,178 = 22/4, YFP-BKα 761–1,178 + full-length RBP2 = 20/4. Statistical significance was calculated with Student's *t*-test (**$P < 0.01$, ***$P < 0.001$).

To confirm the presence of a direct interaction between RBPs and BKα, we produced recombinant GST-fused fragments of RBP1 (aa 771–1,086) and RBP2 (aa 298–604) containing only the FN3 domains (Appendix Fig S4A and B). We found that these fragments were sufficient to successfully pull-down YFP-BKα (aa761–1,178) containing the RCK2 domain from HEK293T cell lysates (Appendix Fig S4C and D). The presence of this interaction was further strengthened by co-immunoprecipitation experiments from transfected HEK293T cells co-expressing full-length RBP2 with $Ca^{2+}$-channels (Cav2.2) and BKα-channels (Appendix Fig S5). Viewed together, these experiments indicate that the two RBP isoforms expressed in the brain (RBP1 and RBP2) can bind directly and simultaneously to $Ca^{2+}$-channels via their SH3-domains and to BKα-channels via their FN3-domains.

**RBP2 alters the voltage dependence of BKα-channels expressed in HEK293T cells**

All experiments up to this point showing that RBPs bind to BKα were performed without regard to the channel function of BKα. To independently test whether RBP2 binds to functional BKα-channels in a membranous environment, we examined the effects of co-expressing full-length RBP2 or defined fragments of RBP2 on BKα-mediated $K^+$-currents in transfected HEK293T cells (Fig 4A). In these experiments, we omitted the different auxiliary subunits of BK-channels because investigation of the relationship of the multiple auxiliary BK-subunits to RBP2 would have exceeded the scope of the current study.

HEK293T cells expressing BKα-channels exhibited large non-inactivating outward currents induced by voltage steps up to +110 mV in the presence of an intracellular $Ca^{2+}$-concentration of 10 μM (Fig 4B, black trace). Co-expression of BKα-channel with both full-length RBP2 (Fig 4B, red trace) and a truncated version of RBP2 containing the FN3-domains and the linker region (Fig 4B,

specific for this domain or if RBP2 can also bind to the RCK1-domain. Using co-immunoprecipitations (Fig 3A and B) and imaging experiments (Fig 3C and D), we found that full-length RBP2 binds to both RCK-domains of BKα-channels, consistent with the similar structure of these domains (Appendix Fig S3B). Similar results were found when the immunoprecipitations were performed with the truncated version of RBP2 containing only the FN3-domains and adjacent linker sequences (aa 247–859) (Appendix Fig S3C).

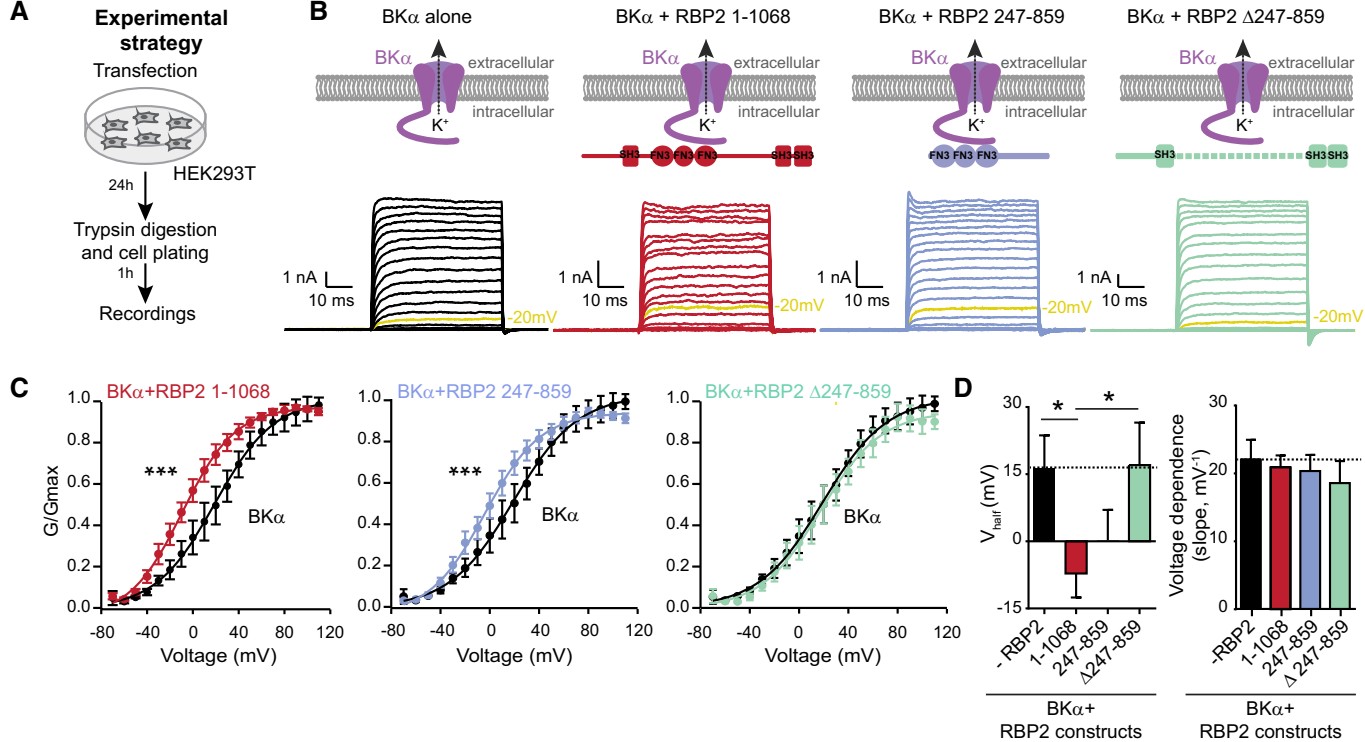

**Figure 4. RBP2 shifts the voltage dependence of BKα-channels expressed in HEK293T cells.**

A  Schematic of the experimental strategy for analysing the effects of various RBP2 proteins on BKα-mediated currents in HEK293T.

B  Diagram of expressed proteins (top) and representative traces of BK-currents recorded in HEK293T in response to 10 mV step depolarizations from −80 mV to +110 mV (bottom, note that in the representative traces currents evoked by −20 mV depolarization are shown in yellow). BKα-currents were recorded in the absence or presence of full-length RBP2 [residues 1–1,068], or of RBP2 fragment containing [residues 247–859] or lacking [Δ247–859] the FN3-domains.

C  Summary plots of the BKα-channel conductance as a function of voltage. Relative conductances (G/$G_{max}$, where $G_{max}$ is the maximal conductance to control for differences in BKα expression levels) were used instead of current measurements to account for the contribution of the driving force and were calculated as G = I/(V−$V_{rev}$), with $V_{rev}$ = RT/zF ln([K]out/[K]in) [here $V_{rev}$ = −80.8203 mV]. Data were fitted with a Boltzmann equation.

D  Bar graphs of the half-maximal activation voltage ($V_{half}$, left) and the slope of the conductance/voltage relation of BKα-channels (right) based on Boltzmann fits of the data in (C).

Data information: Data in (C and D) are means ± SEM; n (cells): BKα only = 10, full-length RBP2 = 10, RBP2 247–859 = 14, and RBP2 Δ247–859 = 6. Statistical significance was assessed using two-way ANOVA for repetitive measurements (C) or one-way ANOVA (D), both followed by Bonferroni *post hoc* test (**P < 0.05, ***P < 0.001).

blue trace) caused a shift in the voltage dependence of BKα-channels, whereas co-expression of a deleted version of RBP2 in which the FN3-domains and the linker region were removed had no effect (Fig 4B, green trace). Quantification of the conductance–voltage (G-V) relationships for BKα-mediated currents under the various conditions demonstrated that the observed changes were significant (Fig 4C). Specifically, RBP2 variants that bound to BKα-protein *in vitro* shifted the voltage dependence of the BKα-current activation by 15–20 mV without affecting the slope of the voltage dependence, whereas the RBP2 variant that did not bind to BKα-protein *in vitro* had no effect (Fig 4D). Therefore, these experiments confirm with a functional assay that the interaction between BKα-channels and RBPs requires the FN3-domains of RBPs.

**Deletion of RBPs decreases presynaptic BK-currents**

To test whether the interaction between RBPs and BKα-channels is physiologically significant, we turned to the calyx of Held, a giant

synapse in the auditory system that allows direct access to the presynaptic terminal (Schneggenburger & Forsythe, 2006). Because presynaptic terminals of the calyx of Held synapse can be patched, this synapse enables direct measurements of various presynaptic ion currents, revealing among others prominent presynaptic BK-currents (Ishikawa *et al*, 2003; Nakamura & Takahashi, 2007). We deleted RBPs in presynaptic calyx of Held terminals by crossing RBP1,2 conditional double KO mice with Krox-20 Cre-driver mice that express Cre-recombinase in presynaptic neurons of the calyx of Held synapse (Voiculescu *et al*, 2000; Acuna *et al*, 2015; Fig 5A). Since only RBP1 and RBP2 but not RBP3 are detectable in the brain, the double deletion of RBP1 and RBP2 completely removes RBPs from calyx synapses. In the following, we analysed littermate RBP1,2 conditional double KO mice either lacking (referred to as "WT") or expressing Cre-recombinase (referred to as "DKO").

We prepared acute brainstem slices from P10-12 WT and DKO mice, and patched presynaptic calyx of Held terminals (Fig 5B). To monitor BK-currents, we blocked voltage-gated Na$^+$- and

**A**    **Breeding strategy**

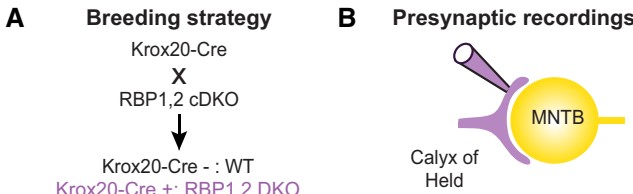

**B**    **Presynaptic recordings**

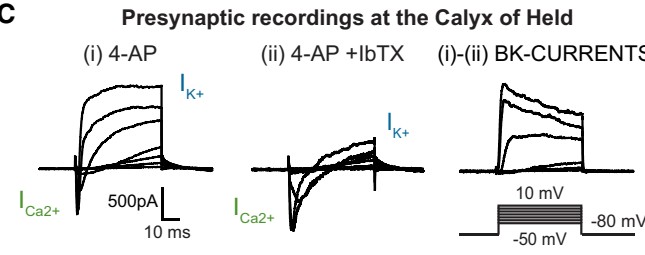

**C**    **Presynaptic recordings at the Calyx of Held**

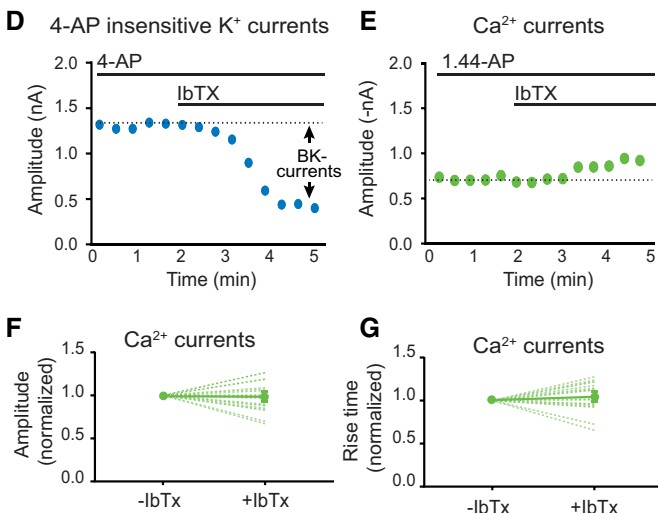

**Figure 5. Experimental approach to recording presynaptic K+-currents mediated by BK-channels at the calyx of Held synapse.**

A   Breeding strategy to obtain mice with a conditional deletion of RBP1 and RBP2 at the calyx of Held synapse using Krox-20 Cre-driver mice (Voiculescu *et al*, 2000). Matings of RBP1,2 conditional DKO (cDKO) mice containing or lacking Krox-20 Cre alleles produced littermate DKO and WT mice.

B   Schematic of whole-cell patch-clamp recordings from presynaptic calyx of Held terminals.

C   Presynaptic currents recorded in response to stepwise 50-ms depolarizations of presynaptic terminals from −50 to +10 mV (10 mV increments) in the presence of tetrodotoxin (TTX; 1 μM) and 4-aminopyridine (4-AP; 2.5 mM), without (i) and with (ii) addition of 0.2 μM iberiotoxin (IbTx). BK-currents are calculated by subtracting (ii) from (i) as IbTx-sensitive K+-currents. Note that in these recordings, inward $Ca^{2+}$-currents ($I_{Ca}^{2+}$) and outward K+-currents ($I_K^+$) are monitored, of which only BK-currents are IbTx-sensitive.

D   Time course of 4-AP-insensitive K+-currents as recorded in a representative experiment as a function of the application of iberiotoxin (IbTx, 0.2 μM).

E   Same as (D), but for $Ca^{2+}$-currents to illustrate that IbTx has no effect on $Ca^{2+}$-channels.

F   Summary plot of the $Ca^{2+}$-current amplitude before and after treatment with IbTX. Single cell values are represented with dotted lines, while the average is shown by full line.

G   Same as (F) but for $Ca^{2+}$-current rise times.

Data information: Data are means ± SEM. Significance was calculated using Student's *t*-test; *n* = 19; *P* > 0.05.

K+-channels with tetrodotoxin (TTX) and 4-aminopyridine (4-AP), respectively, and measured currents in response to progressive step depolarizations with 10 mV intervals (Fig 5C; Nakamura & Takahashi, 2007). Under these conditions, high-voltage activated $Ca^{2+}$-currents manifested as inward currents with a rapid onset, and 4-AP-insensitive K+-currents manifested as outward currents with a delayed onset (Fig 5C). 4-AP-insensitive K+-currents are comprised of BK-currents and of other slowly activating K+-currents (Nakamura & Takahashi, 2007). Subsequent bath application of iberiotoxin (IbTx, 0.2 μM), a specific inhibitor of BK-channels (Galvez *et al*, 1990), suppressed the majority of the 4-AP-insensitive K+-currents that correspond to BK-currents (Fig 5C, compare traces (i) and (ii); quantified in Fig 5D). Treatment with IbTx retained a low-amplitude 4-AP-insensitive K+-current that is probably due to TEA-sensitive K+-conductances, and had no effect on $Ca^{2+}$-currents (Fig 5C and E–G). Overall, these data indicate that most of the 4-AP-insensitive K+-currents in the presynaptic calyx terminal are due to BK-currents.

Using this protocol, we first analysed 4-AP-insensitive K+-currents in WT and DKO calyx of Held terminals (Fig 6). We found that the density of the 4-AP-insensitive K+-current was significantly decreased in RBP-deficient terminals depolarized to 10 mV, and that the time constant of the fast component of the current was increased more than fivefold (Fig 6A–C). The remaining 4-AP-insensitive K+-current after addition of IbTx, however, was not impaired in RBP-deficient presynaptic terminals (Fig 6D–F).

We then calculated the BK-current for each terminal by subtracting the IbTx-insensitive K+-current from the total 4-AP-insensitive K+-current (Fig 6G–M). Strikingly, the BK-current density was significantly reduced in RBP-deficient presynaptic terminals compared to WT terminals from littermate controls (Fig 6G–I). Moreover, we found that deletion of RBP1,2 rendered no change in the voltage dependence of the BK-current activation (Fig 6J), but induced a large increase in the rise times of BK-currents (i.e., their activation kinetics, Fig 6L) and caused a delay in BK-current onset as reflected in an increased latency (Fig 6M). Note that the nerve terminal capacitance was unaltered, suggesting that these changes are not due to a difference in the architecture of the terminal (Fig 6N).

Viewed together, these results suggest that deletion of RBPs from the presynaptic calyx terminal causes a decrease in the concentration of BK-channels. However, the gating of BK-channels is dependent on $Ca^{2+}$, which in turn depends on the activity of voltage-gated $Ca^{2+}$-channels in the presynaptic calyx of Held terminals (Berkefeld & Fakler, 2013). Therefore, it is possible that the decrease in BK-currents observed in the RBP-deficient terminals is not due to an impairment in BK-channels, but caused by a decrease in voltage-gated $Ca^{2+}$-influx. We tested this possibility directly and found that, consistent with previous results (Acuna *et al*, 2015), the $Ca^{2+}$-current density and rise times (as a measure of activation kinetics) were not significantly different between WT and RBP1,2 DKO calyx of Held terminals (Appendix Fig S6). This result shows that the phenotype of reduced BK-currents we observed in RBP-deficient presynaptic terminals is not due to a reduction in the presynaptic $Ca^{2+}$-influx.

Since the RBP DKO causes a loss of tight coupling of $Ca^{2+}$-channels to the release machinery even though it does not produce a loss of $Ca^{2+}$-currents as such (Acuna *et al*, 2015), it is conceivable that the decrease in BK-currents in RBP1,2 DKO terminals is an indirect effect of this function of RBPs. To test this remote possibility, we measured BK-currents in control and RBP1,2 DKO terminals in the

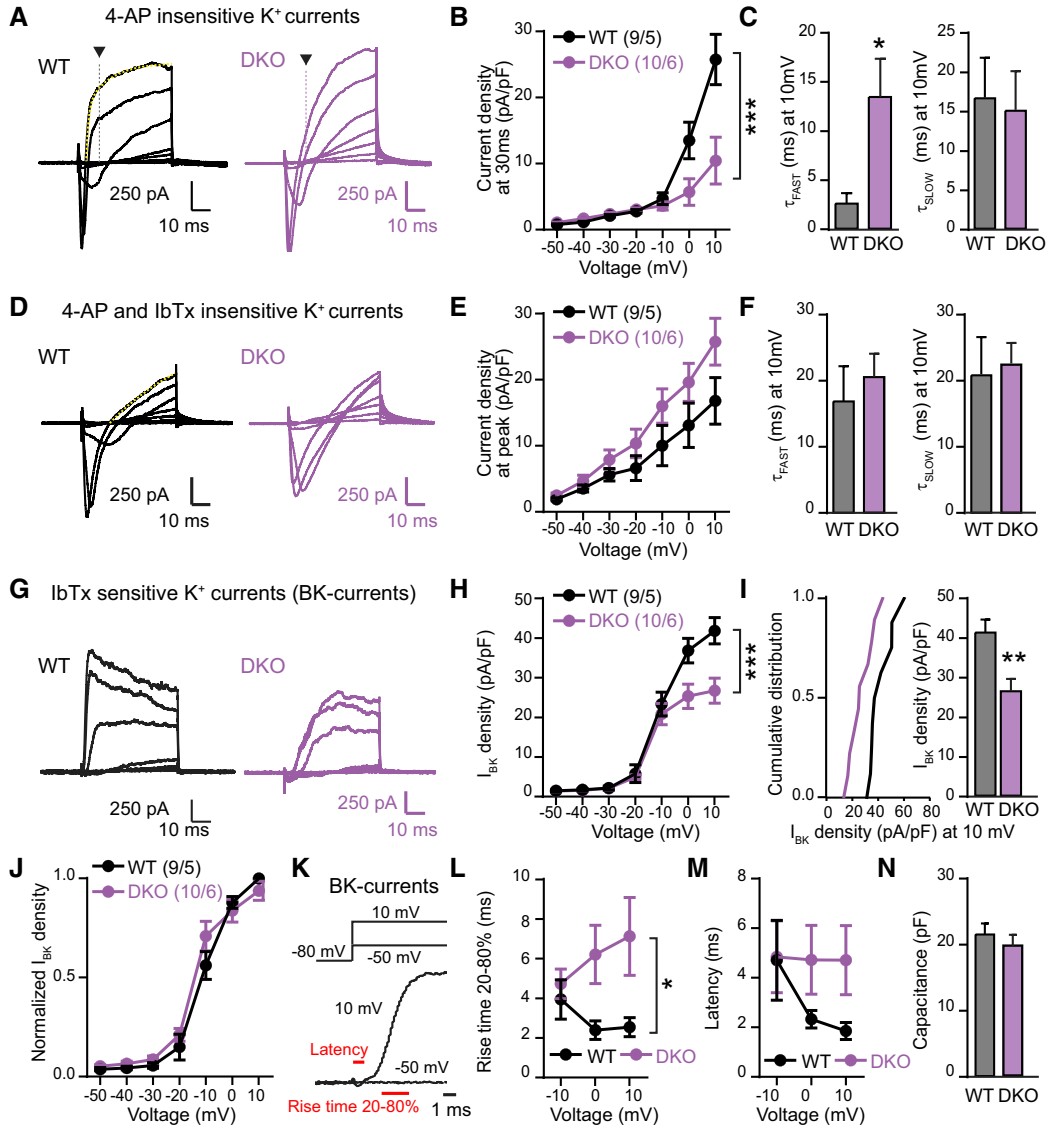

**Figure 6. Deletion of RBPs decreases and decelerates presynaptic BK-currents.**

All experiments were performed by patch-clamp recordings from presynaptic calyx of Held terminals from littermate RBP1,2 conditional DKO mice that lacked (WT) or expressed Krox-20 Cre (DKO) at P10-12. Low concentration of EGTA (0.2 mM) in the internal solution was used for these experiments.

A Sample traces of $Ca^{2+}$-currents and 4-AP-insensitive $K^+$-currents induced by step depolarizations (from −50 mV to +10 mV in 10 mV increments, monitored in 2.5 mM 4-AP). Arrows indicate 30-ms time points, used for analysis in (B). The yellow dotted line superimposed to the trace represents fitting of the curve with a double exponential function, for analysis in (C).

B Summary plot of the mean density of 4-AP-insensitive $K^+$-currents as a function of voltage measured at 30 ms after depolarization as described in (A).

C Summary graphs of the fast ($\tau_{fast}$, left) and slow time constants ($\tau_{slow}$, right) of 4-AP-insensitive $K^+$-currents monitored at +10 mV depolarization. Currents were fitted with a double exponential equation to calculate time constants.

D–F Same as (A–C), but for currents measured after additional application of iberiotoxin (IbTx, 0.2 μM). Current density was measured at the peak.

G Sample traces of BK-currents calculated by subtracting currents produced by depolarizations in the presence of both iberiotoxin and 4-AP from currents produced in the presence of only 4-AP for a given calyx terminal.

H Summary plots of the mean density of BK-currents in presynaptic calyx of Held terminals, BK-current density is plotted as a function of the depolarization voltage, measured at the peak amplitude.

I Cumulative distribution plot (left) and mean value at 10 mV (right) of the BK-current density as described in (H).

J Same as (H), but normalized to the peak current to demonstrate that the deletion of RBPs has no effect on the voltage dependence of BK-channel activation.

K Sample traces for BK-currents monitored at −50 mV and +10 mV to illustrate measurements of 20–80% rise times and latencies of BK-currents.

L Summary plot of mean 20–80% rise times of BK-channels in presynaptic calyx of Held terminals monitored at −10 mV, 0 mV, and +10 mV.

M Summary plot of mean latencies of BK-channels analysed as in (L).

N Summary graph of the capacitance of calyx of Held terminals.

Data information: Data are means ± SEM. Significance was calculated using Student's *t*-test (panels C, F, I, and N) or two-way ANOVA for repetitive measurements followed by Bonferroni *post hoc* test (panels B, E, H, J, L, and M); *n* = 9 terminals/5 mice [WT] or 10 terminals/6 mice [DKO]; *$P < 0.05$, **$P < 0.01$, ***$P < 0.001$. For $Ca^{2+}$-current quantifications, see Appendix Fig S6.

 

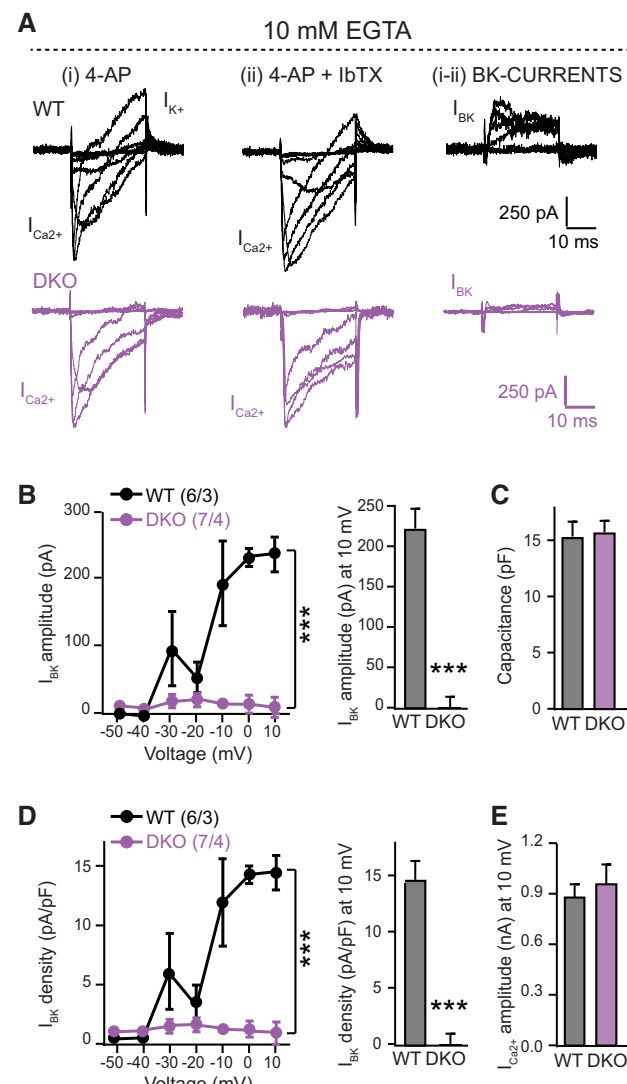

**Figure 7. BK-currents from RBP1,2 DKO mice are completely blocked by high concentration of EGTA (10 mM).**

All experiments were performed by patch-clamp recordings from presynaptic calyx of Held terminals in the presence of high concentration of EGTA (10 mM). Littermate RBP1,2 conditional DKO mice that lacked (WT) or expressed Krox-20 Cre (DKO) at P10-12 were used for these experiments.

A   Sample traces of presynaptic currents recorded in response to stepwise 50-ms depolarizations from −50 to +10 mV (10 mV increments) in the presence of tetrodotoxin (TTX; 1 μM) and 4-aminopyridine (4-AP; 2.5 mM), without (i) and with (ii) addition of 0.2 μM iberiotoxin (IbTx). BK-currents are calculated by subtracting (ii) from (i) as IbTx-sensitive K$^+$-currents (right panel).

B   Summary plots of the mean maximal amplitude of BK-currents in presynaptic calyx of Held terminals from WT and DKO mice as a function of the depolarization voltage. The mean value of BK current amplitude at +10 mV was plotted in the histogram on the right.

C   Summary graph of the capacitance of calyx of Held terminals.

D   Same as (B), but plotting the BK current density.

E   Summary plots of the mean amplitude of Ca$^{2+}$ currents [inward currents analysed from trace (i)].

Data information: Data are means ± SEM. Significance was calculated using Student's *t*-test (graphs in B–E) or two-way ANOVA for repetitive measurements followed by Bonferroni *post hoc* test (I–V plots in B and D); *n* = 6 terminals/3 mice [WT] or 7 terminals/4 mice [DKO]; ***P < 0.001.

presence of high concentration of the slow Ca$^{2+}$-chelator EGTA (10 mM) in the patch pipette. Moreover, with this experiment, we further explored whether the changes in gating kinetics and latency of BK-currents observed in RBP1,2 DKO calyces were due to alterations in the coupling within voltage-gated Ca$^{2+}$-channels and BK-channels. EGTA is a Ca$^{2+}$-chelator with slow Ca$^{2+}$-binding rate constant that has been used to selectively interfere with processes that require Ca$^{2+}$-microdomains, and therefore are loosely coupled to Ca$^{2+}$-channels (at distances higher than 50 nm from Ca$^{2+}$-channels). However, processes that require Ca$^{2+}$-nano-domains and are placed at a distance within 20–50 nm from the Ca$^{2+}$ source are not sensitive to EGTA, but can be blocked by fast Ca$^{2+}$-chelators, such as BAPTA (Fakler & Adelman, 2008; Acuna *et al*, 2015; Luo *et al*, 2017). We found that at the calyx of Held, application of high concentration of EGTA (10 mM) completely blocks BK-currents in RBP1,2 DKO mice, confirming that removal of RBPs affects the coupling of these channels to voltage-gated Ca$^{2+}$-channels (Fig 7A, B and D). Somewhat surprisingly, BK-currents in WT mice were also decreased by EGTA, suggesting that at the age tested (P10-12), a fraction of BK-channels is only loosely coupled to Ca$^{2+}$-channels. As expected from previous results, no differences in the capacitance (Fig 7C) and the Ca$^{2+}$-currents (Fig 7E) were observed in WT and RBP1,2 DKO mice, suggesting that the phenotype was specific for BK-channels.

These experiments let us to conclude that removal of RBPs contributes the correct localization of BK-channels at the presynaptic terminal, controlling their coupling to voltage-gated calcium channels.

### RBPs do not affect expression, but impair presynaptic localization of BKα-channel protein

The reduction in BK-currents we observed upon deletion of RBPs in the calyx of Held (Fig 6G–I) could potentially be explained by changes in the expression levels or by a loss of presynaptic BKα-channels, i.e., a mislocalization of BKα-channels away from the active zone.

In order to examine whether removal of RBPs interferes with the expression of BK-channels, we performed immunoblotting analyses. We prepared total brain lysates from WT and constitutive RBP1,2 DKO mice and analysed the protein levels of the different subunits of BK-channels (Fig 8A). BK-channels are formed by tetramerization of pore-forming BKα-subunits that recruit auxiliary BKβ-subunits (β1–β4; Atkinson *et al*, 1991; Adelman *et al*, 1992; Butler *et al*, 1993; Tseng-Crank *et al*, 1996; Behrens *et al*, 2000; Brenner *et al*, 2000; Uebele *et al*, 2000; Poulsen *et al*, 2009). In the brain, BKβ2 and BKβ4 subunits are most abundant, while BKβ3 is undetectable (Poulsen *et al*, 2009). We found that the RBP1,2 DKO had no effect on the levels of BKα, BKβ1, BKβ2, and BKβ4 as measured by quantitative immunoblotting of whole-brain homogenates (Fig 8A), suggesting that RBPs are not regulating the expression levels of BK-channels or their auxiliary subunits. Similarly, the ventral cochlear nucleus (VCN), which contains the cell bodies of the neurons forming calyx synapses, exhibited no significant differences in the level of BKα and BKβ subunits (Appendix Fig S7B). These data support the idea that genetic removal of RBPs does not affect the overall expression of BK channel subunits. Interestingly, when we performed immunoblotting in the micro-dissected medial nucleus of the trapezoid body (MNTB) containing calyx synapses, we found a trend towards reduced BKα levels and a significant decrease in the

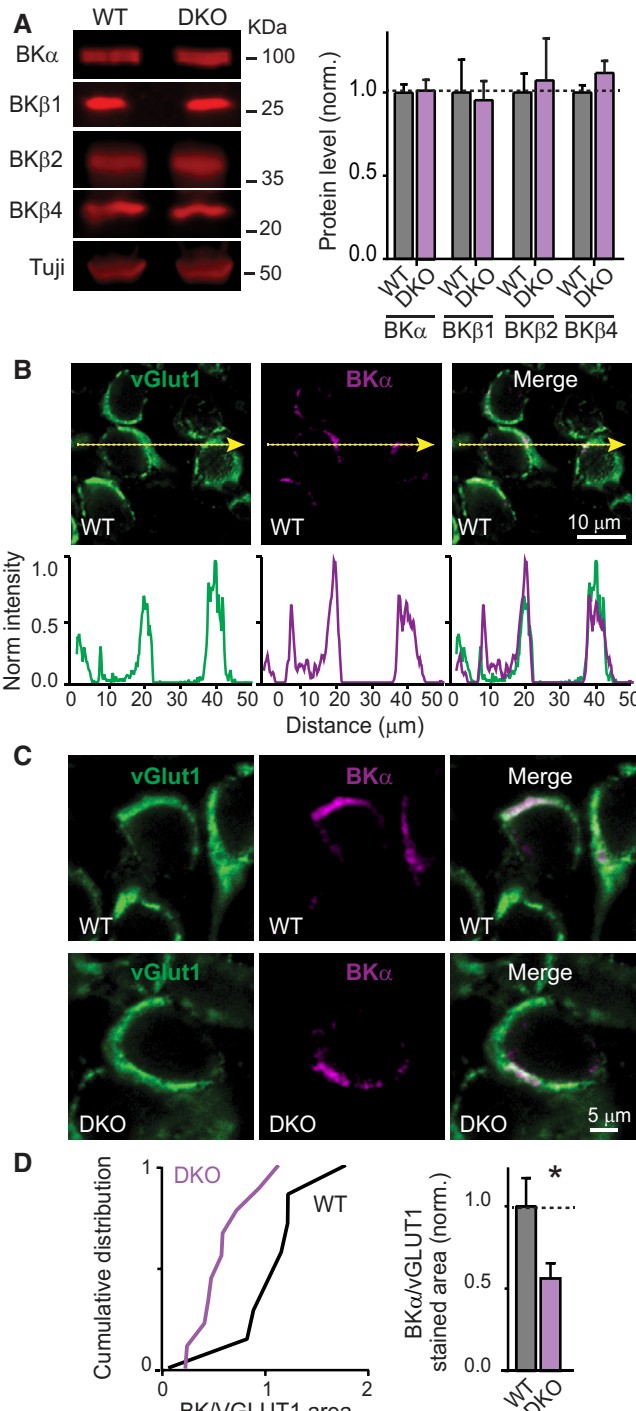

**Figure 8.   Deletion of RBPs causes partial loss of BKα-channels from presynaptic terminals without altering BKα expression levels.**

A   Immunoblot analysis of the levels of BKα-, BKβ1-, BKβ2-, and BKβ4-subunits of BK-channels in total brain homogenates from littermate WT and constitutive RBP1,2 DKO mice (left, representative blots; right, summary graph of levels normalized to tubulin (Tuji) and to WT levels; *n* = 6 WT and DKO mice).

B   Illustration of the strategy for quantitative immunofluorescence analyses of the presynaptic localization of vGlut1 (green) and BKα (magenta) in the calyx of Held synapse in control mice (top, representative images [dashed arrows = position of optical sections for quantitative analyses]; bottom, normalized profiles of the vGlut1 and BKα staining intensity in the calyx of Held terminals as revealed by optical sections).

C   Immunofluorescence measurements of the BKα levels in presynaptic terminals of calyx of Held synapses in littermate RBP1,2 conditional DKO mice that lacked (WT) or expressed Krox-20 Cre (DKO). Brainstem sections of the medial nucleus of the trapezoid body (MNTB) containing calyx synapses were stained for vGlut1 (green) and BKα-channels (magenta), and BKα levels were determined using optical sections as illustrated in (B). BKα staining area was normalized for that of vGluT1 as an internal standard in the same sections to control for differences in staining efficiencies and section plane.

D   Summary graph of the mean presynaptic BKα staining area normalized to the vGluT1 staining area as described in (C) (*n* = 8 WT and 9 DKO mice, averaging multiple images from each mouse as described in the Appendix).

Data information: Data in (A and D) are means ± SEM. Significance was calculated using Student's *t*-test (*$P < 0.05$). See Appendix Fig S7 for more details

of Held synapse, BKα-channels partially co-localized with vGlut1, confirming their presynaptic localization (Fig 8B). In RBP-deficient DKO mice, the presynaptic area containing BKα-channels at the calyx of Held synapse was significantly decreased (~45%; Fig 8C and D). These results confirm that RBPs are necessary to ensure the correct localization of BK-channels at the presynaptic terminal.

## Discussion

Presynaptic BK-channels fine-tune the temporal pattern of APs, thereby controlling the dynamics of neurotransmitter release from nerve terminals (Storm, 1987; Hu *et al*, 2001; Fakler & Adelman, 2008). In the brain, BK-channels are largely present in a complex with voltage-gated $Ca^{2+}$-channels that are also partly presynaptic, thus enabling tight coupling of $Ca^{2+}$-influx to both $Ca^{2+}$-dependent exocytosis and $Ca^{2+}$-dependent BK-channel activation (Roberts, 1993; Robitaille *et al*, 1993a; Knaus *et al*, 1996; Zhou *et al*, 1999; Hu *et al*, 2001; Ishikawa *et al*, 2003; Misonou *et al*, 2006; Nakamura & Takahashi, 2007). How BK-channels are recruited into a complex with $Ca^{2+}$-channels to presynaptic release sites, however, was unknown. Presynaptic release sites are formed by active zones, which are composed of large complexes of multi-domain proteins (Sudhof, 2012). RBPs are a component of active zones that, among others, control the fidelity of neurotransmitter release by enabling tight coupling of voltage-gated $Ca^{2+}$-influx to $Ca^{2+}$-triggered synaptic vesicle exocytosis (Acuna *et al*, 2015, 2016; Grauel *et al*, 2016; Krinner *et al*, 2017; Luo *et al*, 2017). In addition, RBPs redundantly support the essential functions of RIMs, their eponymous binding partners, in the tethering and priming of synaptic vesicles prior to $Ca^{2+}$-triggered exocytosis (Acuna *et al*, 2016). RBPs contain two major domains, SH3- and FN3-domains, consistent with a scaffold that operates by simultaneous binding of

level of BKβ4, while the other subunits were unchanged (Appendix Fig S7C). In the MNTB, we also observed no changes in the level of synaptotagmin-2 (Syt2) and parvalbumin (PV), used as markers for the calyx of Held synapses.

The immunoblotting data suggest that deletion of RBPs may lead to a specific loss of BK-channels from calyx of Held terminals. To further explore this possibility, we measured presynaptic BKα-protein levels by immunofluorescence, using the presynaptic marker vGlut1 as an internal control. We found that at the wild-type calyx

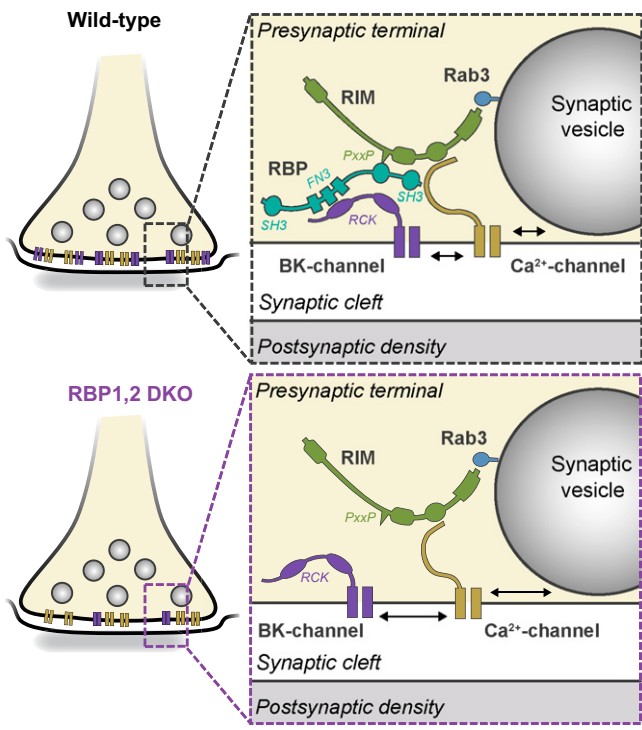

**Figure 9. Cartoon model of the function of RBPs in recruiting BK- and Ca²⁺-channels to presynaptic active zones.**

In wild-type presynaptic terminals (left), RBPs are proposed to simultaneously bind to BK-channels, Ca²⁺-channels, and RIMs, thereby enabling tight coupling of Ca²⁺-influx to both Ca²⁺-triggering of neurotransmitter release via Ca²⁺-binding to synaptotagmins and Ca²⁺-activation of BK-channels via Ca²⁺-binding to the RCK-domains of BKα. Deletion of RBPs leads to a partial loss of presynaptic BK-channels from the terminal, and an increase in the distance of Ca²⁺-channels to the release machinery. As a result, deletion of RBPs causes a decrease in presynaptic BK-currents and an impairment in the tight coupling of Ca²⁺-influx to release during AP trains.

multiple interactors. Multiple binding partners for RBPs are known, including RIMs, voltage-gated Ca²⁺-channels, and bassoon (Wang *et al*, 2000; Hibino *et al*, 2002; Kaeser *et al*, 2011; Davydova *et al*, 2014). All of these proteins bind to the SH3-domains of RBPs, however, and no ligands for the FN3-domains of RBPs were known. To search for potential interactors for the RBP FN3-domains, we here performed an unbiased yeast two-hybrid screen that identified BKα-channels as candidate binding partners for RBPs (Fig 1, Appendix Fig S1, Table S1 and S2). We found that BKα-channels physically interact with RBPs (Figs 2–4) and that deletion of RBPs leads to a decrease in BKα-currents in presynaptic terminals (Figs 5–7). Thus, we propose that BKα-channels are recruited as a component of the macromolecular active zone complex by direct binding to RBPs, which in turn bind to RIMs and to voltage-gated Ca²⁺-channels (Fig 9).

The evidence in support of this proposal can be summarized as follows: First, BKα-channels were found to directly bind to RBPs by four different protein interaction measurements, namely yeast two-hybrid assays (Fig 1A, Appendix Fig S1), co-immunoprecipitations (using both expression systems and native brain homogenates, Fig 1, Appendix Fig S2), imaging of co-expressed proteins (Fig 1D and E), and pull-down experiments (Appendix Fig S4). Each

individual measurement may be considered inconclusive owing to the limitations of the underlying methods, but in combination these measurements provide strong evidence for an interaction. Second, mapping of binding with defined BKα- and RBP2-domains revealed that the RCK-domains of BKα bound specifically to RBP2 sequences containing its FN3-domains, whereas the SH3-domains were dispensable (Figs 2 and 3). This conclusion was further supported by pull-down experiments using recombinant FN3-domains of both RBP1 and RBP2 (Appendix Fig S4). Third, co-expression of RBP2 sequences that contained the BKα-binding site but not co-expression of RBP2 sequences that lacked the BKα-binding impacted the function of BKα-channels reconstituted in HEK293T cells (Fig 4). We interpret these results purely as evidence for a specific interaction, but not as indicative of a particular function of this interaction since the conditions of the HEK293T cell expression of the BKα-channel are too artificial for the experiment to be functionally meaningful. Fourth, deletion of RBPs from the calyx of Held synapse produced a significant reduction in both BK-current amplitude and kinetics, demonstrating that endogenous RBPs are essential for full recruitment of BK-channels to presynaptic terminals (Figs 6 and 7). Fifth, immunofluorescence showed that deletion of RBPs from the calyx of Held synapse caused a physical loss of BKα-channel protein from the presynaptic nerve terminal, which was confirmed by immunoblotting analyses of micro-dissected MNTB samples (Fig 8, Appendix Fig S7).

Although we believe that these data strongly support the proposal that RBPs serve to recruit BK-channels to presynaptic active zones in proximity to Ca²⁺-channels (Fig 9), several questions may be raised about the interpretation of these data. Is it possible that the loss of presynaptic BKα-channels caused by the RBP1,2 DKO in the calyx of Held synapse is an indirect consequence of the mislocalization of voltage-gated Ca²⁺-channels (Acuna *et al*, 2015; Grauel *et al*, 2016)? This possibility seems highly unlikely because deletion of RBPs in DKO synapses by itself does not cause a change in total presynaptic Ca²⁺-currents, but only a modest uncoupling of release from APs during high-frequency stimulus trains (Appendix Fig S5; Acuna *et al*, 2015). Thus, this possibility would not explain the physical loss of BKα-channels from presynaptic terminals. Another question is whether our results obtained at the unusually large calyx of Held synapse are generally applicable to all synapses. Although we cannot rule out that other mechanisms guide recruitment of BK-channels to presynaptic terminals in other synapses, this possibility again appears unlikely given that the function of RBPs operates at all synapses (Acuna *et al*, 2015; Grauel *et al*, 2016; Krinner *et al*, 2017; Luo *et al*, 2017), and that in total brain, RBPs appear to be present in a complex both with BKα-channels and with Ca²⁺-channels (Fig 1, Appendix Fig S2). Yet another question regards the relation of RBP binding to BKα-channels to the interaction of BKα-channels with BKβ-subunits. Although *in vivo* BK-channels operate likely as a multi-subunit complex containing BKα and auxiliary BKβ-subunits, our *in vitro* studies were performed in the absence of BKβ-subunits. Thus, we do not know whether BKβ-subunits affect RBP binding to BKα-channels either positively or negatively, a subject that requires extensive additional studies. However, the RBP1,2 DKO phenotype reflects a change in the entire BK-complex, not just BKα-channels alone, suggesting that RBP clearly is essential for the proper localization of the entire complex.

In summary, we here propose that BKα-channels are recruited into the active zone complex by a direct interaction with RBPs, which also bind to voltage-gated $Ca^{2+}$-channels and to RIMs as central active zone components (Fig 9). As a result, the active zone protein complex is much more extensive as previously envisioned and includes not only $Ca^{2+}$-channels and cell adhesion molecules that are coupled to the synaptic vesicle fusion machinery (Sudhof, 2012), but also with BKα-channels proteins that limit the extent of membrane depolarization during an AP and thereby the amount of neurotransmitter release. This molecular architecture of the active zone complex is likely to have significant implications for the regulation of release under diverse physiological and pathological conditions.

# Materials and Methods

All experiments were performed on anonymized coded samples. For detailed experimental procedures, see Appendix. All animal procedures conformed to National Institutes of Health Guidelines for the Care and Use of Laboratory Animals and were approved by the Stanford University Administrative Panel on Laboratory Animal Care.

### Yeast two-hybrid experiments

Yeast two-hybrid screens of a rat cDNA library were performed as described (Wang *et al*, 1997; Kaeser *et al*, 2011; see Appendix for details). Three bait vectors expressing: (i) truncated RBP2 containing residues 247–859, encoding the FN3-domains and interspersed sequences; (ii) truncated RBP2 containing residues 1–859, encoding the N-terminal SH-domain and the FN3-domains of BP2; and (iii) truncated RBP2 containing residues 247–1,068, encoding the FN3-domains and the C-terminal SH3-domains. All preys were validated by re-assessing their interaction in yeast two-hybrid assays.

### Immunoprecipitations and immunoblotting experiments

Protein interactions were validated by immunoprecipitations of proteins expressed in HEK293T cells. Proteins were co-expressed in HEK293T cells in various combinations as specified in the figures and figure legends by co-transfection. Immunoprecipitation and immunoblotting experiments were performed using standard procedures (see Appendix). For analyses of endogenous brain proteins, experiments used brain homogenates solubilized in Triton X-100 from control and RBP1/RBP2 constitutive DKO mice (Acuna *et al*, 2015).

### Analyses of the RBP interaction with BKα using imaging of transfected HEK293T cells

HEK293T cells were co-transfected with constructs encoding various RBP2, BKα, and YFP proteins as described in the figures and figure legends, and analysed by immunocytochemistry and imaging 24 h later. Protein localizations were quantified in line scans of the fluorescence signal of a cell by measuring the correlation of nuclear DAPI signals with signals emitted by YFP or YFP-tagged RCK-domains from BKα.

### RBP1 and RBP2 double-mutant mice

RBP1/RBP2 conditional and constitutive DKO mice were described previously (Acuna *et al*, 2015, 2016). To delete RBP1 and RBP2 in the calyx of Held, conditional DKO mice were crossed with Krox-20-Cre mice (Voiculescu *et al*, 2000).

### Electrophysiology

Presynaptic BK-currents were recorded in whole-cell voltage clamp configuration from calyx of Held terminals. The internal solution contained (in mM): 97.5 potassium gluconate, 32.5 KCl, 10 HEPES, 1 $MgCl_2$, 12 $Na_2$ phosphocreatine, 2 ATP-Mg, and 0.5 GTP-Na (295–305 mosmol/l, pH 7.3 adjusted with KOH; final $K^+$ concentration, 143.5 mM). The internal solution was supplemented with the calcium chelator EGTA at the concentration of 0.2 mM (for Figs 5 and 6) and 10 mM (for Fig 7). The external solution (standard ACSF) contained (mM): 125 NaCl, 2.5 KCl, 25 $NaHCO_3$, 1.25 $NaH_2PO_4$, 2 $CaCl_2$, 1 $MgCl_2$, 10 glucose, 3 myo-inositol, 2 sodium pyruvate, and 0.5 ascorbic acid, pH 7.4 when bubbled with 95% $O_2$ and 5% $CO_2$. Tetrodotoxin (TTX; 1 μM) and 4-aminopyridine (4-AP; 2.5 mM) were added to the external solution to block, respectively, $Na^+$ and $K^+$ channels. BK-channels were blocked by application of Iberiotoxin (IbTX; 200 nM) to ACSF, perfusing slices at the rate of 1.0 ml/min.

### Immunohistochemistry of brainstem sections

Immunohistochemistry experiments were performed as previously described (Zhang *et al*, 2015) (see Appendix). Serial confocal z-stack images were acquired using a Nikon confocal microscope (A1Rsi). Immunofluorescence was analysed with Nikon analysis software.

**Expanded View** for this article is available online.

## Acknowledgements

We thank Dr. Dick Wu and Dr. Taulant Bacaj for helpful discussion and technical advice. This work was supported by a grant from NIMH (MH086403).

## Author contributions

CA performed the mouse genetics and electrophysiological experiments; FL performed the electrophysiological recordings with 10 mM EGTA; AS performed all other experiments; AS, CA, FL and TCS planned the experiments, analysed data, and wrote the paper.

## Conflict of interest

The authors declare that they have no conflict of interest.

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
