## [Review Process File · The EMBO Journal]

RIM-Binding Proteins Recruit BK-Channels to Presynaptic Release Sites Adjacent to Voltage-Gated Ca²⁺-Channels

Alessandra Sclip, Claudio Acuna, Fujun Luo, and Thomas C. Südhof

Review timeline:

Submission date:	16 November 2017
Editorial Decision:	23 December 2017
Revision received:	11 February 2018
Editorial Decision:	6 April 2018
Revision received:	24 May 2018
Editorial Decision:	28 May 2018
Revision received:	1 June 2018
Accepted:	4 June 2018

Editor: Daniel Klimmeck

Transaction Report:

1st Editorial Decision

23 December 2017

Thank you for the submission of your manuscript (EMBOJ-2017-98637) to The EMBO Journal. Your study has been sent to three referees, and we have received reports from all of them, which I copy below.

As you will see, the referees acknowledge the potential high interest and novelty of your work, although they also express a number of issues that will have to be addressed before they can support publication of your manuscript in The EMBO Journal. In more detail, while referee #3 is overall more positive on the work, referee #1 is concerned about discrepancies of your current findings with earlier reports and lack of proof for endogeneous RBP2 - BKα interactions (ref#1, pt. (i)). In addition, this referee states that your claims on BK dependence of RBP- voltage-gated Ca²⁺ channel interactions are not sufficiently well supported by the data, and points to the in his/her view unresolved cell-type context (ref#1, pt. (ii)). Further, referee #2 is concerned about lack of proof for RBP-dependence of BK - Ca²⁺ channel coupling (ref#2, pt. 5). In addition, the referees list a number of issues related to technical documentation, data illustration and missing controls, which need to be addressed to achieve the level of robustness required for The EMBO Journal.

I judge the comments of the referees to be generally reasonable and we are in principle happy to invite you to revise your manuscript experimentally to address the referees' comments. Please note however, that we would need strong support from the referees on such a revised version of the manuscript to move towards publication. I agree that it would be essential to corroborate the physiological relevance of RBP2, BK and voltage-gated Ca²⁺ channel interdependencies in the presynaptic compartment, as these aspects are at the core of the proposed model.

 Referee #1:

Report on ms EMBOJ-2017-98637 by Scip et al.

The work investigates structural and functional coupling between voltage-dependent Ca²⁺ channels (VDCC) and Ca²⁺-activated large-conductance potassium channels (BK). Starting out from a yeast-2HA screen, the authors show interaction between defined domains of the Rim Binding Protein 2 (RBP2) and the BK alpha-subunit. Based on subsequent experiments involving biochemistry and electrophysiological recordings this interaction is proposed as a mechanism for the tight functional coupling between VDCCs and BK channels in the presynaptic compartment of neurons.

The author's suggestion is in contrast to previous work demonstrating direct association of BK channels with a subset of VDCCs mediated through their respective pore-forming subunits (e.g. Berkefeld et al., *Science*, 2006; Loan et al., *J Cell Sci*, 2007).

Major points:

(i) While interaction of selected domains of RBP2 and BK, and interaction (of both proteins) in heterologous expression experiments appears convincing (Figs. 1-3), recapitulation of the RBP2-BK interaction in the context of native proteins (as present in neurons) failed (Fig. 1G). Thus, BK channels were neither effectively dissolved (on contrast to results by e.g. Berkefeld et al. or Loan et al.), nor were they enriched significantly in RBP2 co-IPs. Also, there are no proper negative controls shown for these IPs.

(ii) The recordings on presynaptic terminals of wildtype and RBP1,2 (double) knockout animals are critical both with respect to experimental conditions and the authors' interpretation of functional coupling in Ca²⁺ nano-domains.

First, as no Ca²⁺ buffering was used, the (high number of) VDCCs effectively activate BK channels even if they are not partnering with each other (through direct structural coupling). Without differential buffering (e.g. high/low concentrations of EGTA and BAPTA), meaningful conclusions can hardly be established.

Second, deletion of RBP2 was shown to result in re-organization of the VDCCs (Grauel et al., 2016), and thus should lead to changes in the time-course of BK activation (Fig. 6) without supporting the inferred RBP-mediated association of VDCCs and BK. Consequently, localization and co-localization of VDCCs and BKs should be performed to clarify this point(s).

(iii) The authors finding of 'RBPs are not regulating the expression levels of BK-channels or their auxiliary subunits' based on immunoblots of BK channel levels in total brain homogenates (Fig. 7) may not hold for the cell-type specific knocked-out in brainstem neurons (used for functional recordings).

Minor points:

- Fig. 2C shows that there is a certain degree of membrane co-localization of BK and RBP2 also in the absence of RBP2-FN3 domains, raising the question if FN3 domains are really important for RBP2-binding to BK channels.

- Fig. 4B, the current traces shown are not meaningful, as alterations in gating parameters are not visible and, in fact, the changes shown in panel C are minor (and hardly convincing).

- Fig. 6J, in patch-clamp recordings from Calyx terminals there is no change in voltage-dependence of BK activation between WT and RBP1,2 conditional KO mice. However, RBP2 induces a left shift in the BK activation curve when expressed in HEK cells. Is there an explanation for this discrepancy?

- Fig. 6K, there is no time scale indicated (although this is the decisive parameter).

- In the 'Method' section for electrophysiology in HEK cells:

How was intracellular Ca²⁺ of 10 μM achieved (again without any buffering or control measurements provided)?

Based on the considerations above, I am afraid I cannot recommend publication of this work in the present form; in fact, a number of additional experiments would be necessary to support the authors conclusions.

Referee #2:

The speed and precision of Ca²⁺ mediated exocytosis at the presynaptic active zone is a well-studied topic and yet there are missing links to complete the picture. The authors here provide a piece to the puzzle: BK-channels. Ca²⁺ activated, with large K⁺ conductance, BK-channels regulate the duration of action potential (AP), thereby monitoring the Ca²⁺ release and hence the neurotransmitter release. Known to be present at the presynapse, here the authors propose a mode of its integration at the AZ as a part of the RIM-RBP- Ca²⁺ channel complex that is instrumental in regulation of exocytosis.

Through yeast two-hybrid assays, co-IP and imaging in expression system like HEK cells the authors provide evidence for the interaction of FN3 domains of RBPs with the RCK-domains of BK-channels. Through electrophysiological analysis in HEK cells and in RBP1,2 DKO calyx of Held authors demonstrate how in the absence of RBPs, the BK-mediated K⁺ current and the abundance of BK channels is impaired at the presynaptic release sites.

In all the authors propose that RBPs directly interact with BK-channels to recruit them to the AZ as a part of the multi-domain protein complex centered on RIM and engaging Ca²⁺ channels, synaptic vesicles and the exocytotic machinery.

- specific major concerns essential to be addressed to support the conclusions

1, In the results section (Page 9) the slice electrophysiology is referred to as *in vivo*. Please correct it to *ex vivo*. It seems inaccurate to refer to electrophysiology done in slices at 22-25°C as *in vivo*.

2, Fig 5 and Fig 6: In the main text (Page 10) a point is made about the application of IbTx not affecting Ca²⁺ currents. Please include a graph illustrating the Ca²⁺ current amplitudes and kinetics.

3, Fig 6: in A the current density is calculated by considering current amplitude at 30ms after depolarization. However, in D and G the peak current is considered. Why not consider the peak current for all, as even in 6A the rise of current is slower?

4, Fig 7: D- in the text (Page 11, last paragraph, 6th line) 'In RBP-deficient DKO mice, the presynaptic area containing BK....' Here a point is made that the presynaptic area where the BK channels are present is reduced but in the legend of Figure 7D it states that 'the mean presynaptic BK α staining intensity' is being compared. Either it is a misunderstanding about how the measurement is done or there is an error in reporting the result. Either way could you please clarify what was measured, the area coverage of the channels or the intensity and how was it measured?

5, Fig 8 and the accompanying text: There is sufficient evidence provided that in the absence of RBP even though the expression level of BK channels is not changed, their presynaptic localization is impaired and they are 'free to diffuse out of the active zone'. However, while the BK channels are presumed to be in nanoscale coupling with Ca²⁺ channels at the active zone in physiology, there is no direct evidence provided for an increase in coupling distance between BK-channels and Ca²⁺ channels in the absence of RBP. Hence it would be advisable to refrain from suggesting so in Figure 8.

6. Referencing previous work is generally balanced, but some relevant work was missed: I suggest to quote the intricate functional coupling of Cav and BK channels as shown by several classical presynaptic studies (e.g. Roberts 1993). Moreover, when quoting the role of RBP at active zones, a recent hair cell ribbon synapse paper (Krinner et al., 2017) should be considered, in which careful optical single active analysis demonstrated a reduction of the number of presynaptic Ca channels in the absence of RBP.

7. The authors should caution the reader about the caveats of experiments using overexpression in heterologous systems.

- minor concerns that should be addressed

1, Introduction Page 3, Line 3: Leitz and Kavalali 2016. Perhaps the authors can include a better reference for synaptic vesicle exocytosis, as this review was targeted to mainly address synaptic vesicle endocytosis.

2. end of first section suggest change to "how they are molecularly connected to"

3, Introduction Page 3, Line 12 (towards discussion): 'BK channels ...controlling extent of Ca²⁺

channel opening...'. However in the absence of RBP, the BK channel localization to active zones and the BK current is impaired, yet there is no change in Ca channel opening (activation/inactivation from Acuna 2015) and Ca current kinetics. Perhaps the authors can discuss this briefly in context of their previous publication Acuna et. al. 2015.

4, Introduction Page 4 Paragraph 2: 'As a result, at these synapses...' Here the references are all bundled together at the end of the sentences. It might be better to provide each reference next its accompanying observation.

5, Figure 1A legend: 'Diagram of RBP2 baits...' The said diagram is only present in the supplementary figure. So either it should be included in the main figure or the reference to it be removed from the main figure legend.

6, Results Page 7 Paragraph 1 and legend of Figure 1D: 'YFP and YFP-tagged BK α RCK2-domain is partly nuclear in the absence of RBP2...' Here please change to 'YFP and YFP-tagged BK α RCK2-domain in the absence of RBP2 is partly nuclear...' since the former sentence gives the impression that YFP is also partly nuclear in the absence of RBP as opposed to YFP being nuclear regardless of presence or absence of RBP2

7, Figure S2, S4: There is no IB in the figure but only in the legend.

8, Fig6: for A and D where the Tau is compared, could you please show an example fit?

9, Discussion Page 13: The third point about BK channels and RBP2 Co-IP, since it is the same as the first point, could they not be integrated together?

10, Discussion Page 13: 5th-7th points please include the accompanying figure no in parentheses as done for previous points.

11, perhaps the age of the animals (P10-12) can be included in the main text.

12, Figure 3B and accompanying main text: Would it be possible to check for the binding with RCK domains of BK channels using the FN3 fragment of RBP2 instead of the full length RBP2, as in the previous figure FN3 fragment was found to be the region of binding in RBP2 to BK channels?

Referee #3:

I find the manuscript convincing, and of high experimental standards. The results are important, and are presented appropriately.

The only suggestion that may increase further the appeal of this manuscript would be to complement Figures 7 or 8 with images from immunostainings of BK channels and RBP in synapses, taken at super-resolution (STED, STORM, expansion microscopy, or any related techniques). This would convince the reader even further. However, this is a minor revision, which is not essential for the publication of the manuscript.

1st Revision - authors' response

11 February 2018

Alessandra Scip et al., "RIM-Binding Proteins Recruit BK-Channels to Presynaptic Release Sites Adjacent to Voltage-Gated Ca²⁺-Channels"

We thank the reviewers and editors for their careful consideration of our paper. We are now resubmitting a revised version of our paper that was amended in view of the reviewers' and editors' comments as described in detail below.

As an overall comment, we would like to emphasize the importance of our findings: we are proposing a molecular mechanism for an observation that was reported some time ago, namely the finding that BK channels and Ca²⁺-channels are functionally coupled and physically connected. However, despite compelling evidence for the physiological significance of the BK channel/Ca²⁺-channel complex, no molecular understanding of this complex was available. Please note, as described below, that original pioneering papers from many laboratories, including prominently the Jonas laboratory, did not actually propose a molecular mechanism but only co-immunoprecipitations. Our paper is not in conflict to those original papers – quite the opposite: our paper provides a molecular explanation for the observations in these papers! The physiological significance for the association of BK channels with Ca²⁺-channels that was described in these papers was the motivation for the current study. We propose that BK channels are recruiting to active zone by the same core components of active zones that mediate assembly of primed synaptic

vesicles adjacent to Ca²⁺-channels, and truly believe that by uniting the recruitment of BK channels with the molecular mechanism of active zone assembly, our study reports an advance that will have a long-lasting impact on our understanding of the architecture of presynaptic terminals. We hope the reviewers and editors will concur.

Below, we cite the editors' and reviewers' comments in *italic* typeface, and provide our response in **bold** typeface. We only repeat selected comments by the editors since some of their comments dealt with issues other than the experimental results of our paper, but we list the reviewers' comments in full.

Response to the editor's comments:

"In more detail, while referee #3 is overall more positive on the work, referee #1 is concerned about discrepancies of your current findings with earlier reports and lack of proof for endogenous RBP2 - BKα interactions (ref#1, pt. (i))."

We believe that reviewer #1 may have misunderstood parts of our manuscript, and perhaps the earlier reports as well. Earlier papers, in particular the paper by Berkefeld et al., 2007, demonstrated that BK channels and voltage-dependent calcium channels (VDCCs) form a complex, but did not show that they directly interact. Moreover, they used plasma membrane enriched fractions, which likely contain dendritic/somatic BK channels, whereas pre-synaptic BK channels are the focus of our study. To the best of our knowledge, no previous paper reported data pertaining to the nature of the VDCC/BK channel complex at the level of a mechanistic analysis of domain interactions, and our paper is the first to do so. This is, we believe, what makes our study important – it provides an explanation for a phenomenon that was described in scores on high-profile papers without a molecular mechanism.

We would also like to argue, as described below, that reviewer #1 may not give us appropriate credit for our data supporting an endogenous RBP-BK channel interaction ('proof' is a very strong word – is there ever definitive proof in science?). The reviewer summarily dismisses our data on immunoprecipitations of endogenous proteins (which we have now expanded), possibly because of a misconception about which protein is difficult to solubilize (it is not the BK channel, but RBPs that are insoluble!), but we would like to argue that these data are valid. Moreover, apart from these data we unequivocally demonstrate in our paper that the RBP deletion leads to a loss of BK channels from presynaptic calyx terminals, which is direct *in vivo* evidence for an interaction. We feel that these lines of evidence do deserve consideration, and hope the reviewers and editors will agree.

"In addition, this referee states that your claims on BK dependence of RBP- voltage-gated Ca²⁺ channel interactions are not sufficiently well supported by the data, and points to the in his/her view unresolved cell-type context (ref#1, pt. (ii))."

This must be a misunderstanding – we never suggested that the interaction of RBPs with VDCCs depends on BK channels. In fact, we show the opposite: that the interactions of RBPs with BK channels and VDCCs are mediated by different domains and are independent of each other (see Figure S5).

"Further, referee #2 is concerned about lack of proof for RBP-dependence of BK - Ca²⁺ channel coupling (ref#2, pt. 5)."

Referee #2 states, which we agree with, that "while the BK channels are presumed to be in nanoscale coupling with Ca²⁺ channels at the active zone in physiology, there is no direct evidence provided for an increase in coupling distance between BK-channels and Ca²⁺ channels in the absence of RBP. Hence it would be advisable to refrain from suggesting so in Figure 8". As the reviewer communicates, direct evidence –especially 'proof'– for this coupling, apart from the physiology, is difficult to come by, which is why the reviewer suggests changing the text. We have followed this suggestion.

"In addition, the referees list a number of issues related to technical documentation, data illustration and missing controls, which need to be addressed to achieve the level of robustness required for The EMBO Journal."

We addressed the issues listed by the referees, and provide additional technical documentation and controls as well as new sets of experiments

"I agree that it would be essential to corroborate the physiological relevance of RBP2, BK and

voltage-gated Ca²⁺ channel interdependencies in the presynaptic compartment, as these aspects are at the core of the proposed model.”

We have tried to address this concern as best as we can, and would like to point out that a selective decrease in both BK currents (amplitude and kinetics) and local BK channel protein as a result of genetic RBP deletion is, in our view, a strong evidence for the physiological relevance of their interaction. Note that as discussed above, the overall significance of the coupling of BK- and Ca²⁺-channels was already well established in the literature and is not the aim of the current study.

Response to the reviewers' comments:

Referee #1

Report on ms EMBOJ-2017-98637 by Sclip et al.

The work investigates structural and functional coupling between voltage-dependent Ca²⁺ channels (VDCC) and Ca²⁺-activated large-conductance potassium channels (BK). Starting out from a yeast-2HA screen, the authors show interaction between defined domains of the Rim Binding Protein 2 (RBP2) and the BK alpha-subunit. Based on subsequent experiments involving biochemistry and electrophysiological recordings this interaction is proposed as a mechanism for the tight functional coupling between VDCCs and BK channels in the presynaptic compartment of neurons.

The authors suggestion is in contrast to previous work demonstrating direct association of BK channels with a subset of VDCCs mediated through their respective pore-forming subunits (e.g. Berkefeld et al., Science, 2006; Loan et al., J Cell Sci, 2007).

We appreciate the reviewer's assessment of our manuscript, but we do not think our paper is in conflict with previous publications. We would like to clarify that these previous reports show that VDCCs and BK channels can be co-isolated by immunoprecipitations both in membrane fractions from brain and in expression systems. We think those papers nicely show that BK and VGCC form a macromolecular complex, but do not believe that they demonstrated “direct association of BK channels with a subset of VDCCs”, as suggested by the reviewer. In fact, the existing data allow no conclusion about the nature (direct or indirect) of the association of VDCCs with BK channels. Thus, we respectfully disagree with the notion that previous studies are in conflict with the results presented in our paper.

Major points:

(i) While interaction of selected domains of RBP2 and BK, and interaction (of both proteins) in heterologous expression experiments appears convincing (Figs. 1-3), recapitulation of the RBP2-BK interaction in the context of native proteins (as present in neurons) failed (Fig. 1G). Thus, BK channels were neither effectively dissolved (on contrast to results by e.g. Berkefeld et al. or Loan et al.), nor were they enriched significantly in RBP2 co-IPs. Also, there are no proper negative controls shown for these IPs.

We are happy to hear the reviewer thinks our in vitro experiments convincingly show the domain-specific interaction between RBPs and BK. However, we are a bit puzzled as to why the reviewer writes “recapitulation of the RBP2-BK interaction in the context of native proteins (as present in neurons) failed”. The problem of analyzing active zone proteins by immunoprecipitations is legendary – it is not the solubilization of BK channels that is difficult, but that of RBPs and active zone proteins. Nevertheless, the fact remains that despite this well documented problem, we do observe co-immunoprecipitation. Moreover, we would like to point out that the extent of enrichment in our experiments cannot be compared to the one presented in the Berkefeld and Loane papers: in their experiments, these authors used plasma membrane enriched fractions, which likely contain dendritic/somatic and not presynaptic BK channels, while we tried to isolate the presynaptic pool, by performing immunoprecipitations with an antibody against RBP2, which is mostly localized to active zones.

Nevertheless, to provide further evidence for the role of RBPs in the BK channel/Ca²⁺-channel complex, we provide a set of additional controls for the immunoprecipitations in the revised paper. The overall controls we use are as follows. First, in Supplementary Figure 2B we showed that BK channels-RBP co-immunoprecipitation can be disrupted by increasing the concentration of SDS in the buffer, suggesting specificity. Second, we could observe co-immunoprecipitation of BK channel with RIM, a well-known binding partner of RBP, confirming the existence of a macromolecular complex. Third, as a negative control, we showed that other synaptic proteins, such as syntaxin, are not co-immunoprecipitated in the complex. Fourth, as suggested by the reviewer, we have now repeated the

immunoprecipitations in RBP1/2 constitutive knock-out mice, and found no precipitation of BK channels (see Supplementary Figure 2C). Thus, our paper describes multiple lines of evidence (in vitro, in native brain, and functional evidence in brain slices) that support the presence of a direct interaction between RBPs and BK.

(ii) The recordings on presynaptic terminals of wildtype and RBP1,2 (double) knockout animals are critical both with respect to experimental conditions and the authors' interpretation of functional coupling in Ca²⁺ nano-domains.

First, as no Ca²⁺ buffering was used, the (high number of) VDCCs effectively activate BK channels even if they are not partnering with each other (through direct structural coupling).

We agree that the presynaptic calyx recordings from RBP1,2 WT and DKO are critical – this is the exact reason why we performed such experiments. We also agree that in the absence of Ca²⁺-buffers, voltage-dependent Ca²⁺-channels (VDCCs) would effectively activate BK channels. We do not understand, however, the reviewer's suggestion that this activation would be efficient even if VDCCs and BK channels “are not partnering with each other”. In fact, we would argue that the classical papers from the Jonas lab and others show the opposite: that efficient activation of BK channels upon Ca²⁺-entry requires VDCCs located in close proximity to BK channels (a notion perhaps shared by most synaptic biologists!). Based on our results, we propose that presynaptic RBPs indeed help to perform this task of localizing VDCCs in the vicinity of BK channels.

Without differential buffering (e.g. high/low concentrations of EGTA and BAPTA), meaningful conclusions can hardly be established.

Again, we respectfully disagree with the reviewer. We think our experiments in the absence of high concentration of Ca²⁺-buffers are critical, and represent the best evidence for a functional association between RBPs and BK channels. First, we decided not to use ultra-high concentrations of Ca²⁺-buffers because we intended to mimic Ca²⁺-levels under normal condition in a nerve terminal (we used 0.2 mM EGTA). Second, deletion of RBPs under identical buffer conditions impacts both the amplitude and speed of presynaptic BK currents – we cannot think of better experimental evidence (genetic manipulation in vivo plus direct recordings from presynaptic boutons) supporting a role for RBP in mediating the tight coupling between Ca²⁺-channels and BK channels.

Second, deletion of RBP2 was shown to result in re-organization of the VDCCs (Grauel et al., 2016), and thus should lead to changes in the time-course of BK activation (Fig. 6) without supporting the inferred RBP-mediated association of VDCCs and BK. Consequently, localization and co-localization of VDCCs and BKs should be performed to clarify this point(s).

We believe the reviewer may have misunderstood the data in the literature on the role of RBPs in the localization of voltage-dependent Ca²⁺-channels (VDCCs). Grauel et al. (2016) – a paper edited by the senior author of the current submission – did not show a change in organization of VDCCs, but confirmed earlier results we had obtained (Acuna et al., 2015) that coupling of VDCCs to release sites is impaired in the absence of RBPs when high-frequency stimulus trains are examined. The Acuna et al. (2015) paper is more relevant here because it was obtained in slices in the calyx by direct patching, whereas the Grauel et al. (2016) paper is in culture without direct measurements of presynaptic Ca²⁺-currents. The Acuna paper documents unequivocally that in the calyx terminal, the RBP1/2 double knockout does NOT cause a decrease in overall VDCC activity, but selectively impairs the speed of release.

We also don't understand why the reviewer suggests that ‘localization and co-localization of VDCCs and BKs should be performed’. Many previous studies, including the ones cited by the reviewer, established such co-localization. Simply repeating such studies seems unnecessary to us; trying to improve on these studies using super-resolution imaging may be worthwhile and add further details, but would be difficult to do within the scope and time frame of revising a manuscript such as the present paper that already contains a large dataset.

(iii) The authors finding of 'RBPs are not regulating the expression levels of BK-channels or their auxiliary subunits' based on immunoblots of BK channel levels in total brain homogenates (Fig. 7) may not hold for the cell-type specific knocked-out in brainstem neurons (used for functional recordings).

To address this concern, we repeated immunoblotting analyses of BK channel levels in homogenates from the ventral cochlear nucleus (VCN) that contains the cell bodies of

presynaptic neurons forming the calyx of Held synapses, and from the medial nucleus of the trapezoid body (MNTB) that contains the actual calyx synapses (new Supplementary Figure S7). In the VCN, we also found no differences in the expression level of BK α , BK β 1, and BK β 4 subunits, confirming the observation obtained with whole brain homogenates. In the MNTB, we observed a trend towards reduced expression levels of BK α and a significant loss of BK β 4 subunits in DKO mice, while the level of BK β 1 and BK β 2 subunits were unchanged, as well as the level of Synaptotagmin-2 and parvalbumin that were used as markers for the calyx synapses. These data are consistent with the more accurate immunofluorescence data showing a reduction in the presynaptic BK α subunits in the MNTB (Figure 7D).

Minor points:

- Fig. 2C shows that there is a certain degree of membrane co-localization of BK and RBP2 also in the absence of RBP2-FN3 domains, raising the question if FN3 domains are really important for RBP2-binding to BK channels.

As the reviewer stated, a slight increase in the membrane co-localization of BK and RBP2 was observed in the absence of the FN3 domains. However, the change observed is very small and does not reach statistical significance. To further support our conclusions and address the reviewer's concern, we have now performed an additional experiment and could show that recombinant FN3 domains of both RBP1 and RBP2 can pull down the RCK2 domain of BK channel expressed in HEK293T cells, confirming the conclusion that the FN3 domains bind to the RCK2 domain of BK channels (see Figure S4).

- Fig. 4B, the current traces shown are not meaningful, as alterations in gating parameters are not visible and, in fact, the changes shown in panel C are minor (and hardly convincing).

We amended figure 4B and now highlight the -20 mV trace in yellow to help the readers to visualize the gating parameters. However, we respectfully disagree with the reviewer about the significance of the changes in panel C. We could replicate the changes in the presence of the RBP2 fragment containing only the FN3 domains and not in the construct containing the deletion of the FN3 domain, confirming the specificity of the effect.

- Fig. 6J, in patch-clamp recordings from Calyx terminals there is no change in voltage-dependence of BK activation between WT and RBP1,2 conditional KO mice. However, RBP2 induces a left shift in the BK activation curve when expressed in HEK cells. Is there an explanation for this discrepancy?

As explained in detail in the Results and the Discussion section of the paper, we believe that heterologous expression of BK channels in HEK cells provides a tool to probe for interactions, but is too artificial to model the conditions of BK currents in a nerve terminal. We therefore interpreted the effect we observed in HEK cells as evidence for an interaction, not as a demonstration of a physiological effect – this demonstration was achieved by studying the RBP deletion in the calyx. We also want to point out that in the nerve terminal, BK channels associate with auxiliary beta subunits that modify the activation and inactivation properties of the channels and might account for the differences we see.

- Fig. 6K, there is no time scale indicated (although this is the decisive parameter).

We added the time scale as requested by the reviewer

- In the 'Method' section for electrophysiology in HEK cells:

How was intracellular Ca²⁺ of 10 μ M achieved (again without any buffering or control measurements provided)?

A free intracellular Ca²⁺-concentration of ~10 μ M was achieved by mixing Ca²⁺ and Ca²⁺-buffers according to empirical estimations performed in Maxchelator, a resource commonly used by physiologists (<http://maxchelator.stanford.edu/webmaxc/webmaxcE.htm>). As described previously (Berkefeld et al., J. Neurosci. 33, 7358-7367 [2013], Brenner et al, J Biol Chem. 275, 6453-6461 [2000]; Lippiat et al., J Membr. Biol. 192, 141-148 [2003]). Such Ca²⁺-concentration was necessary to achieve reliable activation of BK channels in heterologous systems in the absence of Ca²⁺-channel expression. We added details in the revised supplementary methods describing this point.

Based on the considerations above, I am afraid I cannot recommend publication of this work in the

present form; in fact, a number of additional experiments would be necessary to support the authors conclusions.

We appreciate the reviewer's arguments for rejection of our paper, but note that he/she had no criticism of the two major results of our paper: the effect of the deletion of RBPs on BK currents and on BK localization in a central synapse, and the demonstrated interaction of RBPs with BK channels. The reviewer is asking us to supply additional data whose scope, we believe, exceeds the extent of a regular paper; he/she is asking for these data based on concerns that, as we explained above, may partly reflect a misunderstanding of the literature. Reviewers naturally have a tendency to suggest ever more experiments, but there needs to be a balance between what evidence is really necessary to make a case and what evidence may be more appropriately dispensable. The discovery of a mechanism that explains the coupling of BK channels to VDCCs, a phenomenon that was described by others in a multitude of previous major papers, seems to us to be a significant advance, and we would like to ask the reviewer to consider our arguments, especially the fact that our data do not contradict previous major findings, but instead explain these findings mechanistically.

Referee #2

The speed and precision of Ca²⁺ mediated exocytosis at the presynaptic active zone is a well-studied topic and yet there are missing links to complete the picture. The authors here provide a piece to the puzzle: BK-channels. Ca²⁺ activated, with large K⁺ conductance, BK-channels regulate the duration of action potential (AP), thereby monitoring the Ca²⁺ release and hence the neurotransmitter release. Known to be present at the presynapse, here the authors propose a mode of its integration at the AZ as a part of the RIM-RBP- Ca²⁺ channel complex that is instrumental in regulation of exocytosis.

Through yeast two-hybrid assays, co-IP and imaging in expression system like HEK cells the authors provide evidence for the interaction of FN3 domains of RBPs with the RCK-domains of BK-channels. Through electrophysiological analysis in HEK cells and in RBP1,2 DKO calyx of Held authors demonstrate how in the absence of RBPs, the BK-mediated K⁺ current and the abundance of BK channels is impaired at the presynaptic release sites.

In all the authors propose that RBPs directly interact with BK-channels to recruit them to the AZ as a part of the multi-domain protein complex centered on RIM and engaging Ca²⁺ channels, synaptic vesicles and the exocytotic machinery.

We appreciate this reviewer's impartial assessment of our manuscript, and hope to have addressed her/his concerns satisfactorily as described below.

- specific major concerns essential to be addressed to support the conclusions

1, In the results section (Page 9) the slice electrophysiology is referred to as in vivo. Please correct it to ex vivo. It seems inaccurate to refer to electrophysiology done in slices at 22-250C as in vivo.

Agreed – we corrected the sentence as the reviewer suggested.

2, Fig 5 and Fig 6: In the main text (Page 10) a point is made about the application of IbTx not affecting Ca²⁺ currents. Please include a graph illustrating the Ca²⁺ current amplitudes and kinetics.

Agreed – we added two graphs in figure 5F and 5G illustrating the Ca²⁺ current amplitudes and kinetics before and after the application of Iberitoxin.

3, Fig 6: in A the current density is calculated by considering current amplitude at 30ms after depolarization. However, in D and G the peak current is considered. Why not consider the peak current for all, as even in 6A the rise of current is slower?

In Figure 6A, outward K⁺-currents are comprised of fast activating BK-currents that is sensitive to Iberitoxin but insensitive to 4-AP, and of other slowly activating K⁺-currents that are insensitive to Iberitoxin and 4-AP. The reason why we analysed the amplitude at 30 ms is because at this time point the BK channel mainly contributes to generate the outward currents. We also provide analysis for the 4-AP and Iberitoxin insensitive currents, in Figure 6D-F, and found that they were unchanged. We finally showed that the rise of 4-AP insensitive currents is slower suggesting a decreased in the fast activating BK currents, which was verified in isolated BK currents in Figure 6G-I.

4, Fig 7: D- in the text (Page 11, last paragraph, 6th line) 'In RBP-deficient DKO mice, the

presynaptic area containing BK....' Here a point is made that the presynaptic area where the BK channels are present is reduced but in the legend of Figure 7D it states that 'the mean presynaptic BKα staining intensity' is being compared. Either it is a misunderstanding about how the measurement is done or there is an error in reporting the result. Either way could you please clarify what was measured, the area coverage of the channels or the intensity and how was it measured?

We corrected the discrepancy in the text and added details of the quantification in the extended methods. Briefly, for every section, we calculated the area of presynaptic BK channel colocalizing with vGlut1, using the Nikon analysis software. We then normalized the BK area to the area of the vGlut1 signal to correct for differences in the extension of the MNTB due to different slice plane or distance to bregma. We finally averaged the results obtained from single sections of the same animal, to obtain a graph where the n number is referred to a single animal.

5, Fig 8 and the accompanying text: There is sufficient evidence provided that in the absence of RBP even though the expression level of BK channels is not changed, their presynaptic localization is impaired and they are 'free to diffuse out of the active zone'. However, while the BK channels are presumed to be in nanoscale coupling with Ca²⁺ channels at the active zone in physiology, there is no direct evidence provided for an increase in coupling distance between BK-channels and Ca²⁺ channels in the absence of RBP. Hence it would be advisable to refrain from suggesting so in Figure 8.

Agreed – we have changed the text as suggested.

6. Referencing previous work is generally balanced, but some relevant work was missed: I suggest to quote the intricate functional coupling of Cav and BK channels as shown by several classical presynaptic studies (e.g. Roberts 1993). Moreover, when quoting the role of RBP at active zones, a recent hair cell ribbon synapse paper (Krunner et al., 2017) should be considered, in which careful optical single active analysis demonstrated a reduction of the number of presynaptic Ca channels in the absence of RBP.

Agreed – we added these citations in the manuscript

7. The authors should caution the reader about the caveats of experiments using overexpression in heterologous systems.

Agreed – we now state in the discussion that “The evidence in support of this proposal can be summarized as follows: first, BKα-channels were found to directly bind to RBPs by four different protein-interaction measurements, namely yeast two-hybrid assays, co-immunoprecipitations, imaging of co-expressed proteins, and pull-down experiments (Figure 1, S1, S4). Each individual measurement is inconclusive owing to the limitations of the underlying methods, but in combination they provide strong evidence for an interaction.” To further convince the reviewer, we additionally performed new pull-down experiments with recombinant proteins containing only the FN3 domains (see Figure S4). For this purpose, we purified in bacteria the FN3 domains of both RBP1 and RBP2, fused to GST, and use those to successfully pull down the RCK domain of BK channels expressed in HEK cells. Moreover, because of the inherent caveats of in vitro experiments, we decided to perform extensive genetic experiments, which in our view are the most conclusive ones.

- minor concerns that should be addressed

1, Introduction Page 3, Line 3: Leitz and Kavalali 2016. Perhaps the authors can include a better reference for synaptic vesicle exocytosis, as this review was targeted to mainly address synaptic vesicle endocytosis.

Agreed – done

2. end of first section suggest change to "how they are molecularly connected to"

Agreed – we changed the text as suggested

3, Introduction Page 3, Line 12 (towards discussion): 'BK channels ...controlling extent of Ca²⁺ channel opening....' However in the absence of RBP, the BK channel localization to active zones and the BK current is impaired, yet there is no change in Ca channel opening (activation/inactivation from Acuna 2015) and Ca current kinetics. Perhaps the authors can discuss this briefly in context of their previous publication Acuna et. al. 2015.

Agreed – we now discuss this apparent paradox. We hypothesize that the classical experiments on the role of BK channels controlling the extent of calcium channel opening were performed in reduced systems in which relatively small changes become magnified, whereas in our in vivo analysis in calyx terminals this role of BK channels that depends on their proximity to calcium channels may not be as robust.

4, Introduction Page 4 Paragraph 2: 'As a result, at these synapses...' Here the references are all bundled together at the end of the sentences. It might be better to provide each reference next its accompanying observation.

Agreed – we divided the references

5, Figure 1A legend: 'Diagram of RBP2 baits...' The said diagram is only present in the supplementary figure. So either it should be included in the main figure or the reference to it be removed from the main figure legend.

Agreed – we changed the figure legend accordingly

6, Results Page 7 Paragraph 1 and legend of Figure 1D: 'YFP and YFP-tagged BK α RCK2-domain is partly nuclear in the absence of RBP2...' Here please change to 'YFP and YFP-tagged BK α RCK2-domain in the absence of RBP2 is partly nuclear...' since the former sentence gives the impression that YFP is also partly nuclear in the absence of RBP as opposed to YFP being nuclear regardless of presence or absence of RBP2

Agreed – done

7, Figure S2, S4: There is no IB in the figure but only in the legend.

We corrected the discrepancy

8, Fig6: for A and D where the Tau is compared, could you please show an example fit?

We add an example fit (yellow dotted lines superimposed to the trace) in Figure 6, as requested by the reviewer.

9, Discussion Page 13: The third point about BK channels and RBP2 Co-IP, since it is the same as the first point, could they not be integrated together?

Agreed – done

10, Discussion Page 13: 5th-7th points please include the accompanying figure no in parentheses as done for previous points.

Agreed – done

11, perhaps the age of the animals (P10-12) can be included in the main text.

Agreed – we added the age of the animals in the result section and in the figure legends

12, Figure 3B and accompanying main text: Would it be possible to check for the binding with RCK domains of BK channels using the FN3 fragment of RBP2 instead of the full length RBP2, as in the previous figure FN3 fragment was found to be the region of binding in RBP2 to BK channels?

We added a new experiment confirming the finding of Figure 3B with the FN3 fragment of RBP2 (Figure S3C)

Referee #3

I find the manuscript convincing, and of high experimental standards. The results are important, and are presented appropriately.

The only suggestion that may increase further the appeal of this manuscript would be to complement Figures 7 or 8 with images from immunostainings of BK channels and RBP in synapses, taken at super-resolution (STED, STORM, expansion microscopy, or any related techniques). This would convince the reader even further. However, this is a minor revision, which is not essential for the publication of the manuscript.

We are grateful to this reviewer for the positive and knowledgeable assessment of our paper. We would love to be able to do super-resolution studies of VDCCs and BK channel localizations, but the fact is that such studies have not yet been achieved for BK channels alone, will probably take a multi-year effort, and are clearly beyond the scope of the current manuscript.

Again, we thank the reviewers for their comments, and hope that the revised paper with the changes described above can be deemed acceptable for publication in EMBO J.

2nd Editorial Decision

6 April 2018

Thank you for submitting the revised version of your manuscript and sending us the point-by-point response, as well as your additional comments. Please accept our apologies for the unusual delay in getting back to you due to detailed discussions in the team, which got protracted due to travel. Your revised study has now been re-evaluated by the three referees whose comments are enclosed. As you will see, the first referee remains overall more critical on the study than the two others, however we decided - in light of the strong support of the latter - to give you the opportunity to revise your manuscript to address the referee's points.

Both referee #2 and referee #3 find that their concerns have been sufficiently addressed and are broadly in favour of publication. Referee #1's states persistent concerns regarding insufficient support for the claims made and asks you to consolidate your findings with experiments using different EGTA concentrations.

We do concur with this referee, as well as cross-commenting between the referees, that the mentioned control experiments would be important to rule out ambiguous interpretations of the data. Taking into account the positive comments of referee #2 and referee #3, we have decided that pending a satisfactory revision, we would go ahead with acceptance of this manuscript as soon as possible. Thus, I ask you to revise your manuscript regarding the points raised by referee #1 and evaluate, whether you would be able to add complementary control data, or, alternatively, relativise your statements and introduce caveats where appropriate.

Referee #1:

General notion:

Previous work, in particular using IPs and functional recordings (under high buffer conditions) from heterologous expression systems, showed that formation of macromolecular complexes between some types of voltage-dependent Ca²⁺ channels (VDCC) and Ca²⁺-activated BK channels does not require additional components.

This does, of course, not exclude stabilization of native VDCC-BK complexes by additional proteins such as the RIM-binding proteins (RBPs) that has been shown to abundantly interact with N- and P/Q-type VDCCs in membranes of the mouse/rat brain (see Mueller et al., PNAS, 2010).

My original comments (major and minor points) were essentially meant to substantiate the authors' claim of RBPs being a prerequisite for both the structural and the functional coupling of VDCC and BK channels (as whole proteins - not isolated domains). In this context, I suggested some necessary control experiments for IPs and for recordings from native complexes in the calyx. The result of some of these experiments are included into the revised ms, others were discussed but not performed.

Thus, Western blots of IPs from RBP knockouts were added to the revised supplement (Figure S2); the quality of these blots (high background, section of restricted size), however, precludes reasonable assessment and can hardly be called convincing.

With respect to the functional recordings, the authors added the buffer concentration used for their calyx-experiments (0.2 mM EGTA) to the revised methods section (the respective data for the HEK cell experiments (discussed in the replies) appears to be still missing in the ms).

Unfortunately, the authors did not repeat their calyx-experiments (in WT and RBP KO) at higher buffer conditions (such as 10 mM EGTA), although such conditions would be necessary to discriminate between BK channels co-assembled with VDCCs and 'free' BK channels. Explicitly, large Ca²⁺-currents (as present in the calyx) at low buffer concentrations activate BK channels independent of their integration into VDCC-BK complexes; 0.2 mM EGTA is easily saturated by

the current amplitude shown by the authors.

Taken together, I am still far from being convinced that the authors' claim of RBPs being the 'connecting' element of native VDCC-BK complexes is correct; I certainly agree with the authors' notion that RBPs are important elements for the correct assembly of 'macromolecular active-zone complexes'.

Referee #2:

The authors have satisfied the requests of this reviewer

Referee #3:

My initial review of the manuscript was quite positive, and it remains so after the revisions that the authors performed. I suggest that the manuscript be published.

2nd Revision - authors' response

24 May 2018

Authors' response to the Reviewers' Additional Comments for Alessandra Scipio et al., "RIM-Binding Proteins Recruit BK-Channels to Presynaptic Release Sites Adjacent to Voltage-Gated Ca²⁺-Channels"

We thank the editor and the reviewers for re-evaluating our paper. Below, we cite the editors' and reviewers' comments in *italic* typeface, and provide our response in **bold** typeface. We only repeated selected comments by the editors since some of their comments dealt with issues other than the experimental results of our paper, but we list the reviewers' comments in full.

Response to the editor's comments:

Both referee #2 and referee #3 find that their concerns have been sufficiently addressed and are broadly in favour of publication. Referee #1's states persistent concerns regarding insufficient support for the claims made and asks you to consolidate your findings with experiments using different EGTA concentrations.

We do concur with this referee, as well as cross-commenting between the referees, that the mentioned control experiments would be important to rule out ambiguous interpretations of the data. Taking into account the positive comments of referee #2 and referee #3, we have decided that pending a satisfactory revision, we would go ahead with acceptance of this manuscript as soon as possible. Thus, I ask you to revise your manuscript regarding the points raised by referee #1 and evaluate, whether you would be able to add complementary control data, or, alternatively, relativise your statements and introduce caveats where appropriate.

We are now submitting an updated version of the paper which include experiments performed at the calyx of Held with high concentration of EGTA (10 mM). We included these data as a full figure in the re-revised paper (see new Figure 7). This is discussed in detail below.

Referee #1:

General notion:

Previous work, in particular using IPs and functional recordings (under high buffer conditions) from heterologous expression systems, showed that formation of macromolecular complexes between some types of voltage-dependent Ca²⁺ channels (VDCC) and Ca²⁺-activated BK channels does not require additional components.

This does, of course, not exclude stabilization of native VDCC-BK complexes by additional proteins such as the RIM-binding proteins (RBPs) that has been shown to abundantly interact with N- and P/Q-type VDCCs in membranes of the mouse/rat brain (see Mueller et al., PNAS, 2010).

The reviewer seems to be unhappy about the fact that we may not have given sufficient credit to the excellent paper by Müller et al. on the proteomics of Ca²⁺-channels that was published in 2010 in PNAS. We have no hesitation to acknowledge the outstanding approach of the Müller paper, which in extremely well-controlled experiments revealed a rich array of proteins that are co-immunoprecipitated with Ca²⁺-channels. However, to say that this paper shows that RBPs “abundantly interact with N- and P/Q-type VDCCs in membranes of the mouse/rat brain” may be stretching it a bit. The Müller et al. paper describes the Ca²⁺-channel proteome, but no direct interactions of Ca²⁺-channels. A large number of proteins are co-immunoprecipitated with Ca²⁺-channels, including gratifyingly RBPs that were shown to bind to Ca²⁺-channels a decade earlier by Hudspeth’s group. We apologize if we did not cite the Müller paper sufficiently, but the current paper is not about the binding of RBPs to Ca²⁺-channels, and we did not discuss Ca²⁺-channels extensively in the current paper.

My original comments (major and minor points) were essentially meant to substantiate the authors' claim of RBPs being a prerequisite for both the structural and the functional coupling of VDCC and BK channels (as whole proteins - not isolated domains). In this context, I suggested some necessary control experiments for IPs and for recordings from native complexes in the calyx. The result of some of these experiments are included into the revised ms, others were discussed but not performed.

Thus, Western blots of IPs from RBP knockouts were added to the revised supplement (Figure S2); the quality of these blots (high background, section of restricted size), however, precludes reasonable assessment and can hardly be called convincing.

As explained earlier, IPs of essentially insoluble protein complexes provide a challenge. This is a problem that has continued to make analysis of subsynaptic complexes by IPs difficult. Despite the difficulties related to this experiment, we showed immunoprecipitation of RBPs and BK channels from endogenous brains and the absence of IP in RBP1,2 DKO mice. As requested by the editor, we also provided an uncropped version of the Western blot in Figure S2C.

With respect to the functional recordings, the authors added the buffer concentration used for their calyx-experiments (0.2 mM EGTA) to the revised methods section (the respective data for the HEK cell experiments (discussed in the replies) appears to be still missing in the ms).

We apologize for omitting this information, and added the details of the solutions used for recordings in HEK293T cells in the supplementary methods, as followed:
“The following bath solution was used (in mM): 144 NaCl, 5.8 KCl, 0.9 MgCl₂, 1.3 CaCl₂, 0.1 NaH₂PO₄, 5.6 glucose, 10 HEPES-KOH, 2.5 4-aminopyridine (4-AP) (pH~7.4). The internal solution contained (in mM): 135 KCl, 3.5 MgCl₂, 2 ATPNa₂, 0.5 EGTA, 5.04 CaCl₂, 5 HEPES-NaOH pH 7.2, resulting in a free Ca²⁺-concentration of ~10 μM, which was estimated using the maxchelator website (<http://maxchelator.stanford.edu/webmaxc/webmaxcE.htm>).”

Unfortunately, the authors did not repeat their calyx-experiments (in WT and RBP KO) at higher buffer conditions (such as 10 mM EGTA), although such conditions would be necessary to discriminate between BK channels co-assembled with VDCCs and 'free' BK channels. Explicitly, large Ca²⁺-currents (as present in the calyx) at low buffer concentrations activate BK channels independent of their integration into VDCC-BK complexes; 0.2 mM EGTA is easily saturated by the current amplitude shown by the authors.

Based on the insistence of the reviewer, we have now recruited a new scientist to the project and performed a new series of experiments for the paper that are now shown in the new Figure 7 of the manuscript. In this new set of experiments, we performed Ca²⁺- and BK-current measurements in control and RBP1,2 double KO calyx terminals in the presence of 10 mM EGTA. We found that at the calyx of Held, application of high concentration of EGTA severely decreased BK-currents in control terminals, and completely block BK currents in RBP1,2 DKO mice. The effect of the RBP1,2 DKO was highly significant and confirmed that removal of RBPs dramatically impairs BK-channels (Figure 7A, B, D). Ca²⁺-currents, conversely, were indistinguishable between control and RBP1,2 DKO terminals, as we previously reported (Acuna et al., 2015). It may surprise the reviewer that BK currents were decreased by EGTA in control terminals, suggesting that at the age tested (P10-12) a fraction of BK channels is only loosely coupled to Ca²⁺-channels. However, we believe that given the fact that Ca²⁺-channel tethering to active zones is only modestly dependent on RBPs (see

comments above) and that the various RIM and RBP complexes are likely dynamic, this observation suggests that at least some BK-channels are only loosely coupled to Ca^{2+} -channels in the calyx even if they are tightly coupled to RBPs as we demonstrate. Finally, as expected from previous results, no differences in terminal capacitance were observed (Figure 7C). Overall, these results provide further, by now hopefully overwhelming evidence produced by an independent newly recruited electrophysiologist that RBPs are truly essential for the recruitment of BK-channels to presynaptic terminals.

Taken together, I am still far from being convinced that the authors' claim of RBPs being the 'connecting' element of native VDCC-BK complexes is correct; I certainly agree with the authors notion that RBPs are important elements for the correct assembly of 'macromolecular active-zone complexes'.

We hope that given the latest data repeating the challenging BK-channel measurements in yet another condition will have convinced even this reviewer that RBPs indeed are essential for recruiting BK-channels to the active zone adjacent to voltage-gated Ca^{2+} -channels.

Referee #2:

The authors have satisfied the requests of this reviewer

Referee #3:

My initial review of the manuscript was quite positive, and it remains so after the revisions that the authors performed. I suggest that the manuscript be published.

We thank referees 2 and 3 for the positive comments.

3rd Editorial Decision

28 May 2018

Thank you for submitting the revised version of your manuscript for consideration by The EMBO Journal. I have now evaluated your amended manuscript and concluded that all remaining concerns have been sufficiently addressed.

Thus, we are pleased to inform you that your manuscript has been accepted in principle for publication in The EMBO Journal, pending some minor issues regarding material and methods and formatting as outlined below, which need to be adjusted at re-submission.

Corresponding Author Name: Sclip A, Sudhof TC

Manuscript Number: Manuscript EMBOJ-2017-98637R